# Movie Facts and Fibs (MF²):
# A Benchmark for Long Movie Understanding

## Abstract

Despite recent progress in vision-language models (VLMs), holistic understanding of long-form video content remains a significant challenge, partly due to limitations in current benchmarks. Many focus on peripheral, "needle-in-a-haystack" details, encouraging context-insensitive retrieval over deep comprehension. Others rely on large-scale, semi-automatically generated questions (often produced by language models themselves) that are easier for models to answer but fail to reflect genuine understanding. In this paper, we introduce **MF²**, a new benchmark for evaluating whether models can comprehend, consolidate, and recall key narrative information— requiring integration of both visual and linguistic modalities—from full-length movies (**50-170 minutes long**). MF² includes over 50 full-length, **open-licensed** movies, each paired with manually constructed sets of claim pairs—one true (*fact*) and one plausible but false (*fib*), totalling over 850 pairs. These claims target core narrative elements such as **character motivations** and **emotions**, **causal chains**, and **event order**, and refer to **memorable moments** that humans can recall without rewatching the movie. Instead of multiple-choice formats, we adopt a binary claim evaluation protocol: for each pair, models must correctly identify both the true and false claims. This reduces biases like answer ordering and enables a more precise assessment of reasoning. Our experiments demonstrate that both open-weight and closed state-of-the-art models fall well short of human performance, underscoring the relative ease of the task for humans and their superior ability to retain and reason over critical narrative information—an ability current VLMs lack.

## 1 Introduction

Vision-language models (VLMs) have demonstrated strong performance across a wide range of tasks involving both images and videos (Deitke et al., 2024; Chen et al., 2024b; Liu et al., 2024; Zhang et al., 2024; Bai et al., 2025; Zhang et al., 2025; Xu et al., 2025; Li et al., 2025). As these models continue to scale and improve, a natural next frontier lies in long-form video understanding, essential for real-world applications such as education, storytelling, and other types of narrative video analysis—where success depends on integrating and reasoning over information that unfolds over extended periods.

Despite this progress, current evaluation benchmarks for video understanding remain limited. They often rely on relatively short video content (Lei et al., 2018; Xiao et al., 2021; Wu et al., 2021; Parmar et al., 2024; Rawal et al., 2024; Qiu et al., 2024; Fang et al., 2024) and even when longer videos are available (Huang et al., 2020; Song et al., 2023; Chandrasegaran et al., 2024; Ataallah et al., 2024; Wang et al., 2024b; Fu et al., 2024; Wu et al., 2024), they fail to access genuine comprehension. Instead, many existing benchmarks target "needle-in-a-haystack" retrieval (Kamradt, 2024; Wang et al., 2024a;d; Zhao et al., 2025), focusing on peripheral or low-level details that models can possibly retrieve with long context windows, even without the abstractive understanding of the central storyline that humans use. For example, questions such as *"What color is the liquid inside the bucket in the painting?"* (Wu et al., 2024) or *"Why did Player number 4 in white push down Player number 17 in purple during the match?"* (Wang et al., 2024b) primarily test narrow recall capabilities, rather than engaging with fundamental narrative components. Although some existing benchmarks (Chandrasegaran et al., 2024) do include tasks that touch upon aspects of storyline understanding, they are not primarily designed around narrative-central events. Our benchmark

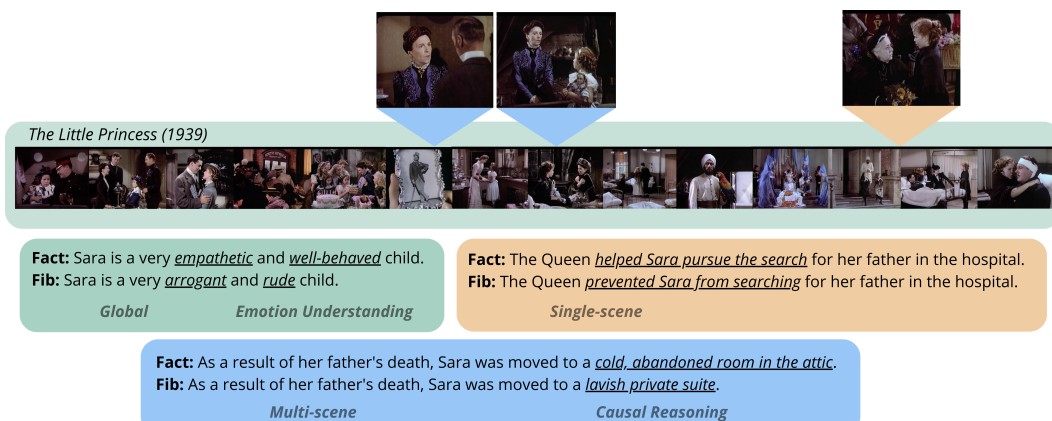

Figure 1: Illustration of three claim pairs (each with a *fact* and a *fib*) from the movie "The Little Princess". Our claims target memorable events, focusing on key turning points of the narrative such as emotional arcs and causal relationships between characters, and require reasoning across different granularities (single-scene, multi-scene and global). More examples are provided in Appendix F.

instead approaches the problem from a different angle: **We argue that referring to memorable moments that humans can recall even without rewatching the movie is key**. Such moments encapsulate critical turning points that shape the narrative trajectory (Papalampidi et al., 2019; 2020), such as emotional arcs or causal relationships between characters and events (see Fig. 1). Other benchmarks prioritize quantity over quality, using semi-automatically generated questions (Chandrasegaran et al., 2024; Ataallah et al., 2024), often produced by language models themselves, which may reflect model biases rather than robust evaluation. Evaluation formats also pose challenges: questions are typically either free-form, making automatic and reliable assessment difficult (Bavaresco et al., 2024; Liu & Zhang, 2025; Ye et al., 2025), or multiple choice-based, suffering from several pitfalls such as answer selection biases based on superficial cues or poorly constructed distractors (Li & Gao, 2024; Loginova et al., 2024; Singh et al., 2025; Molfese et al., 2025). Furthermore, as we highlight in Table 1, access to open-source video content is often restricted due to copyright issues, and even when external links (typically to platforms such as YouTube) are provided, they are prone to becoming inaccessible over time (Wang et al., 2024b), which limits reproducibility and long-term usability. These limitations highlight the need for a fully open-source benchmark that **goes beyond shallow retrieval** and supports rigorous evaluation of narrative understanding.

In this paper, we introduce MF$^2$, a benchmark to evaluate **genuine narrative comprehension** of full-length movies. The dataset comprises 53 full-length, open-licensed movies with an **average duration of 88.33 minutes**. For each movie, we manually construct a set of contrastive claim pairs, each consisting of one true statement (a *fact*) and one plausible but false counterpart (a *fib*). These claim pairs target memorable events in the movie, such as character motivations, causal links, event chronology, and other key aspects that are central to the narrative (see Table 2). Unlike benchmarks that can be solved through brute-force memorization or naïve extensions of context windows (e.g., "needle-in-a-haystack" style queries), MF$^2$ requires models to **consolidate**, **reason**, and **recall** fundamental narrative components across long time spans, requiring integration of both vision and language, and reflecting more human-like understanding. Our contributions are as follows:

1. We present MF$^2$, a benchmark designed for evaluating narrative comprehension of full-length movies. It consists of 53 full-length, open-licensed movies, each accompanied by corresponding subtitles, and includes over 850 human-crafted claim pairs.

2. We shift away from traditional multiple-choice formats and adopt a **contrastive claim evaluation protocol**, following Karpinska et al. (2024): for each contrastive pair, models must correctly identify both the true and false claims, avoiding biases like answer ordering and enabling a more precise reasoning assessment.

3. We perform an extensive evaluation of state-of-the-art open and closed models as well as a human evaluation to establish upper-bound performance, revealing a notable performance gap between models and humans.

Table 1: Comparison of video datasets across different aspects. MC stands for multiple-choice and OE for open-ended questions.

| Dataset | Avg. Duration (mins) | Annotation | Evaluation Format | Source Availability |
|---|---|---|---|---|
| CausalChaos (Parmar et al., 2024) | - | Auto & Manual | MC & OE | Source link not available |
| CinePile (Rawal et al., 2024) | 2.67 | Auto & Manual | MC | YouTube links |
| EgoSchema (Mangalam et al., 2023) | 3.00 | Auto & Manual | MC | Videos |
| ViMuL (Shafique et al., 2025) | 4.52 | Auto & Manual | MC & OE | Videos |
| EgoPlan-Bench2 (Qiu et al., 2024) | up to 5 | Auto & Manual | MC | Videos |
| LongVideoBench (Wu et al., 2024) | 7.89 | Manual | MC | Videos |
| Video-MMMU (Hu et al., 2025b) | 8.44 | Manual | MC | Videos |
| MovieChat-1K (Song et al., 2023) | 9.40 | Manual | MC & OE | Videos |
| MLVU (Zhou et al., 2024) | 12.00 | Auto & Manual | MC & OE | Videos |
| Neptune (Nagrani et al., 2025) | up to 15 | Auto & Manual | MC & OE | Videos |
| Video-MME (Long) (Fu et al., 2024) | 39.76 | Manual | MC | YouTube links |
| HourVideo (Chandrasegaran et al., 2024) | 45.70 | Auto & Manual | MC | Videos |
| InfiniBench (Ataallah et al., 2024) | 52.59 | Auto & Manual | MC & OE | Key frames |
| LVBench (Wang et al., 2024b) | 68.35 | Manual | MC | YouTube links |
| **MF²** | **88.33** | **Manual** | **Claim pairs** | **Videos** |

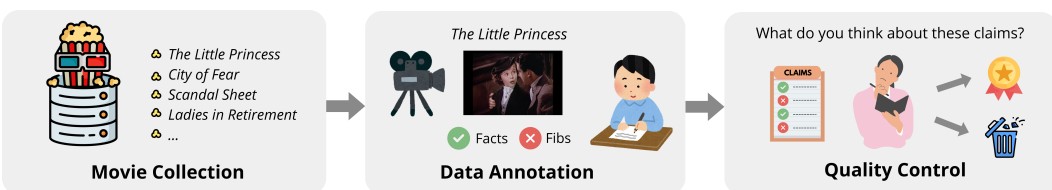

Figure 2: Dataset construction process involving three main stages: movie collection, data annotation, and quality control.

4. We publicly release all data and code[1] to facilitate reproducibility and support future research on long movie understanding.[2]

## 2 MF²: MOVIE FACTS AND FIBS

MF² includes 53 full-length, open-licensed movies, each accompanied by subtitles, and 868 human-authored contrastive claim pairs. Each pair tests whether a model can distinguish true from false information based on its understanding of the story. Fig. 1 shows some examples. We now describe the dataset construction process in detail, covering movie selection (§2.1), annotation methodology including claim categorization and granularity (§2.2), and human quality control procedures used to filter ambiguous or low-quality claims (§2.3). Fig. 2 provides an overview of these three stages.

### 2.1 MOVIE SELECTION AND SUBTITLES

We started by collecting a pool of movies from the Internet Archive,[3] an online repository of open-licensed media. We specifically selected titles released under the Public Domain 1.0 license to ensure legal reusability and support open-access research. To reduce the risk of data contamination in modern foundation models (Jacovi et al., 2023), we focused on older films released between 1920 and 1970, prioritizing those with limited online visibility, measured by the number of user reviews on IMDb. We sourced original-language subtitles—the majority of which are in English—from OpenSubtitles.org,[4] a widely used platform that provides subtitles for a large collection of movies, TV shows, and other video content. For one movie without available subtitles, we used whisper-1 (Radford et al., 2023)[5] to generate a transcript and manually post-edited to ensure high quality. This process yielded a final collection of 53 full-length movies with an average duration of 88.33 minutes,

---

[1] https://anonymous.4open.science/r/MF2

[2] We will release the movies upon acceptance.

[3] https://archive.org

[4] https://www.opensubtitles.org

[5] https://platform.openai.com/docs/models/whisper-1

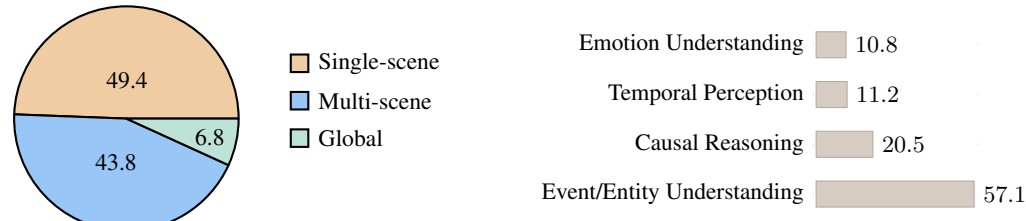

Figure 3: Distribution of claim pairs across reasoning granularities (**left**) and comprehension dimensions (**right**).

each accompanied by audio and aligned subtitles (see §A for details about the movies, including genre, language, and duration).

## 2.2 DATA ANNOTATION

The annotation process involved 26 annotators, all of whom are co-authors of this work, who watched the full movies, identified key narrative elements, and constructed pairs of constrastive claims: one factually correct statement (*fact*) and one minimally altered, false counterpart (*fib*). Following Karpinska et al. (2024), annotators were instructed to minimize lexical differences between the *fact* and the *fib*, changing only the parts needed to flip the truth value. The annotation guidelines are presented in §B. This contrastive formulation serves two purposes: *(i)* it isolates the specific narrative element being tested, reducing the chance that models rely on superficial cues (e.g., sentence length, structure, or other lexical patterns); and *(ii)* it simplifies quality control (see §2.3) by making inconsistencies easier to detect.

**Claim granularity.** To capture different levels of reasoning, annotators labeled each *fact* according to the granularity required to verify its truth: *(i) single-scene*: answerable using information from one scene; *(ii) multi-scene*: requiring integration across multiple scenes; and *(iii) global*: relying on high-level understanding that spans the full movie, including accumulated or inferred information (cannot be easily tied to distinct scenes). As shown in Fig. 3 (left), the dataset includes a balanced distribution of single-scene and multi-scene *facts* (with a smaller proportion requiring global reasoning). Importantly, all claims test long-form comprehension irrespectively of the reasoning granularity: while global claims require reasoning across the entire movie, key events can also unfold within single or multiple scenes. Even single-scene claims are non-trivial, as they assess whether models can extract and retain salient localized information. While humans naturally focus on important elements, models may lack this ability (see §4.2, where we show that this is indeed the case).

**Comprehension dimensions.** In addition to the reasoning granularity, annotators also labeled each claim pair with one or more comprehension dimensions, indicating the specific aspects of narrative understanding being tested. These dimensions, informed by prior work (Xiao et al., 2021; Zhang et al., 2023b; Wang et al., 2024b), are defined in Table 2, with their distribution shown in Fig. 3 (right). Annotators could choose multiple dimensions for the same claim.

## 2.3 QUALITY CONTROL

We conducted a human evaluation stage to establish a human baseline for model comparison (see Section §3), which was also used to collect feedback on the quality of claims. For this round, annotators first selected a subset of movies they had not previously seen during the data annotation stage. After watching a movie, they classified the corresponding claims as either true or false using a custom annotation interface (see §B for an example and full guidelines). Claims were presented one at a time, and annotators were required to respond based solely on memory. To support the identification of problematic claims, we encouraged annotators to leave comments whenever a claim was ambiguous, poorly phrased, open to interpretation, or too fine-grained to be meaningfully tied to narrative understanding (e.g., needle-in-a-haystack claims). The annotation guidelines emphasized the importance of paying close attention while watching the movie, as many claims require subtle

Table 2: Definitions of comprehension dimensions.

| Comprehension Dimension | Definition |
| --- | --- |
| *Event/Entity Understanding* | Involves identifying key entities (e.g., people, places, or objects) and understanding the events they participate in. This includes tracking entities across scenes, interpreting their roles, and recognizing their interactions and relationships throughout the narrative. |
| *Temporal Perception* | Requires reasoning about the timeline of events—determining whether actions occur before, after, or simultaneously—and may also include counting or sequencing events. The focus is on broader temporal relationships within the narrative. |
| *Emotion Understanding* | Involves recognizing the emotional states of characters and interpreting how these emotions evolve throughout the story. |
| *Causal Reasoning* | Focuses on identifying cause-and-effect relationships between events or actions, including both explicit and implicit dependencies that may span multiple scenes. |

reasoning or contextual understanding. Importantly, annotators were instructed not to use any external tools or take notes, ensuring that all responses reflected natural human memory and comprehension.

An optional second stage allowed annotators to revisit their previous responses with access to the movie. This stage was used exclusively to collect additional comments for validation: annotators used it to revise earlier answers after reflecting on the full context of a claim pair.

As part of the filtering process, two annotators reviewed all comments left during the stages described above. Without watching the corresponding movies, and solely based on the comments left, they identified problematic claims and removed them from the dataset. Importantly, no claims were rewritten at this stage—they were either accepted or discarded. This filtering step resulted in the removal of 104 pairs of claims, yielding a cleaner set of 868 high-quality pairs (§A provides more statistics).

## 3 EXPERIMENTAL SETUP

In this section, we describe the setup used to evaluate a range of vision-language models (VLMs) on the $MF^2$ benchmark. Our experiments include both closed and open-weight models, tested across multiple input modalities using a standardized evaluation protocol.

**Modalities.**  We evaluate all models under a vision-language setup, where they receive visual input in the form of sampled movie frames. We also experiment with providing subtitles as additional input. For the ablation studies (see §4.2), we test two other configurations: one that includes movie synopses, and another that provides only the movie title and release year.

**Baselines.**  We experiment with several state-of-the-art vision-language models (VLMs). As closed models, we include GPT-4o (OpenAI et al., 2024) and Gemini 2.5 Pro (Team et al., 2023). Our open-weight models include VideoLLaMA3 (Zhang et al., 2025), Qwen2.5-VL (Bai et al., 2025), LLaVA-Video (Zhang et al., 2024), InternVL3 (Zhu et al., 2025), Ovis2 (Lu et al., 2024), and LongVILA-R1 (Chen et al., 2025), a model specialized for long video benchmarks. For all models except GPT-4o, we first downsample videos to 1 frame per second, following each model's preprocessing approach. From these frames, we then uniformly sample a subset, adjusting the number of frames based on each model's input constraints and original training settings.[6] For GPT-4o, frames are uniformly sampled directly from the original videos without prior downsampling. The exact number of frames sampled per model is reported in Table 3. We test multiple prompt variants and report results using the best-performing prompt for each model. To extract predictions, we use regular expressions to identify

---

[6]Note that models always receive uniformly sampled frames from the full movie—not targeted scene windows. They must process the entire movie and transcript to identify relevant content, irrespectively of the reasoning granularity of the claim.

Table 3: Performance of both open-weight and closed models when evaluated on $MF^2$. We report both pairwise and standard accuracy, when models are assessed on video inputs w/ and w/o subtitles. Best-performing values among models are **bolded** and best for each specific group are underlined.

| Method | #Params | #Frames | Pairwise Accuracy (%) | | Accuracy (%) | |
|---|---|---|---|---|---|---|
| | | | w/o subs | w/ subs | w/o subs | w/ subs |
| *Baselines* | | | | | | |
| Random | - | - | 25.0 | 25.0 | 50.0 | 50.0 |
| Human | - | - | - | 84.1 | - | 90.5 |
| *Closed Models* | | | | | | |
| GPT-4o | - | 50 | 18.8 | 46.8 | 55.2 | 71.4 |
| Gemini 2.5 Pro | - | 120 | **37.2** | **60.6** | **64.2** | **77.6** |
| *Open-weight Models* | | | | | | |
| VideoLLaMA3 | 7B | 180 | 20.5 | 33.5 | 57.0 | 62.7 |
| Qwen2.5-VL | 7B | 180 | 24.6 | 32.8 | 56.7 | 62.0 |
| LLaVA-Video | 7B | 64 | 6.6 | 19.0 | 51.7 | 57.8 |
| LongVILA-R1 | 7B | 180 | 11.5 | 16.9 | 50.1 | 56.6 |
| InternVL3 | 8B | 64 | 10.9 | 36.9 | 53.1 | 64.6 |
| Ovis2 | 34B | 10 | 18.8 | 45.6 | 53.3 | 69.5 |
| Qwen2.5-VL | 72B | 180 | 29.7 | 45.9 | 58.8 | 70.4 |
| LLaVA-Video | 72B | 64 | 15.6 | 41.8 | 54.6 | 69.1 |
| InternVL3 | 78B | 64 | 22.1 | 51.3 | 58.0 | 72.7 |

True/False answers in the model outputs, selecting either the first or last valid match depending on the prompt structure. We include all prompt templates and answer parsing details in §C.1 for reproducibility. We also include a human baseline where evaluators judged claims based on their memory, without rewatching scenes (see §2.3).

**Evaluation protocol.** We report two metrics: *(i)* pairwise accuracy, which measures how often models correctly classify both the true and the false claim in a pair (i.e., they receive credit only if both are labeled correctly; no points are awarded for partial correctness); and *(ii)* standard accuracy, which is computed over individual claims. The random baselines are 25% and 50%, respectively. Following prior work (Karpinska et al., 2024), both models and human annotators see and evaluate each claim independently, without access to the paired structure during prediction (see discussion in §7). Pairwise accuracy is computed post-hoc by grouping predictions from the same pair.

## 4 RESULTS AND ANALYSIS

In this section, we first present the main experimental results (§4.1), followed by ablation studies (§4.2) that analyze model performance across the different input modalities, reasoning granularities, and comprehension dimensions.

### 4.1 MAIN RESULTS

In Table 3, we report both standard and pairwise accuracy for humans, open-weight, and closed models across two input modalities: video-only and video with subtitles. Our results reveal that:

**Both open-weight and closed models fall significantly short of human performance.** Among the closed models, Gemini 2.5 Pro achieves the highest scores, with a pairwise accuracy of 60.6%, followed by the open-weight InternVL3-72B, which performs 9.3% lower, when evaluated on both video and subtitles. Despite their relatively strong performance, both models rank significantly behind humans, with a 24.1% absolute gap. Smaller models perform only marginally above chance, with the best among them exceeding the random baseline by just 11.09%. These findings underscore the difficulty of the task for current models, but also highlight humans' superior ability to retain and reason over critical narrative information.

Table 4: Performance of Gemini 2.5 Pro across different input modalities. *Video* uses only the video stream; *Subs* includes only subtitle information; *Synopsis* relies only on the synopsis of the movie obtained from Wikipedia; *Video w/ Subs* combines both video and subtitles inputs; and *Movie Title* uses only the claim, along with the movie title and release year, without access to movie content.

| | Input Modality | | | | |
|---|---|---|---|---|---|
| **Metric** | **Video** | **Subs** | **Synopsis** | **Video w/ Subs** | **Movie Title** |
| Pairwise Accuracy (%) | 37.2 | 56.7 | 25.5 | 60.6 | 43.7 |
| Accuracy (%) | 64.2 | 76.2 | 61.8 | 77.6 | 66.3 |

**Models, particularly medium and large-sized ones, perform substantially better when subtitles are available compared to relying on video alone.** By contrast, smaller-sized models perform near chance level when evaluated solely on the video and marginally improve with the addition of subtitles. A notable exception is InternVL3-7B, which shows a more pronounced improvement with subtitles, indicating some ability to leverage textual context despite its smaller size. In contrast larger models, such as InternVL3-72B, followed by LLaVA-Video and Ovis2, demonstrate significant gains when subtitles are provided. These results indicate that textual cues can provide meaningful signals when integrated with visual inputs—a dynamic we further explore in the following section, where we deep dive into a fine-grained analysis of different input modalities and reasoning capabilities.

## 4.2 ABLATION ANALYSIS

**Beyond vision: the role of textual and world knowledge.** Table 4 presents an ablation study of Gemini 2.5 Pro, highlighting its strong reliance on subtitles and parametric (internal) knowledge. Notably, the model performs competitively even without visual input. It achieves strong results when provided only with subtitles, or even just the movie title and release year. This suggests that the model draws substantially on broad world knowledge encoded during pretraining. In contrast, performance declines when the model is given only the movie synopsis, indicating that not all forms of textual context are equally helpful. These results underscore the critical role of subtitles as a grounding signal and suggest that pretrained knowledge, rather than surface-level contextual inputs like a synopsis, enables accurate reasoning in the absence of video. Note that these findings deviate somewhat from the general assumption made when providing contextual knowledge; past work steering models to focus on contextual knowledge (e.g. (Li et al., 2023b; Shi et al., 2024; Wang et al., 2025)) or performing retrieval-augmented generation (Lewis et al., 2020) generally assume that the contextual knowledge is correct and contains the correct answer. However, on videos, which represent long and complex contexts, we find that models in fact perform better *without* contextual knowledge.

**Input modality contributions across comprehension dimensions and reasoning granularities.** In Fig. 4, we present ablation studies for Gemini 2.5 Pro, examining how different input modalities contribute to performance across comprehension dimensions and reasoning granularities. We observe that models handle temporal perception more effectively than other comprehension aspects across all modalities—likely because time-related information is often directly observable in visual and textual inputs, making it easier to track and interpret (Zellers et al., 2021; Li et al., 2022). Event and entity understanding is notably weaker under visual-only conditions, likely due to the need for linguistic disambiguation. This limitation becomes evident when subtitles are introduced: the most significant gain is observed in the aforementioned category, highlighting the complementary role of textual context. In contrast, emotional understanding benefits the least from subtitles, indicating challenges in affective comprehension. Beyond comprehension dimensions, reasoning performance under visual-only inputs remains relatively consistent across reasoning types. However, under the presence of textual cues, global reasoning becomes more challenging than single- and multi-scene reasoning.

**A fine-grained view of large-scale model performance across comprehension dimensions and reasoning granularities.** Fig. 5 shows that, among the large-scale models, Gemini 2.5 Pro still demonstrates inferior performance, ranking second to humans in various categories. Other models like LLaVA-Video and InternVL3 generally show lower scores, suggesting areas for improvement.

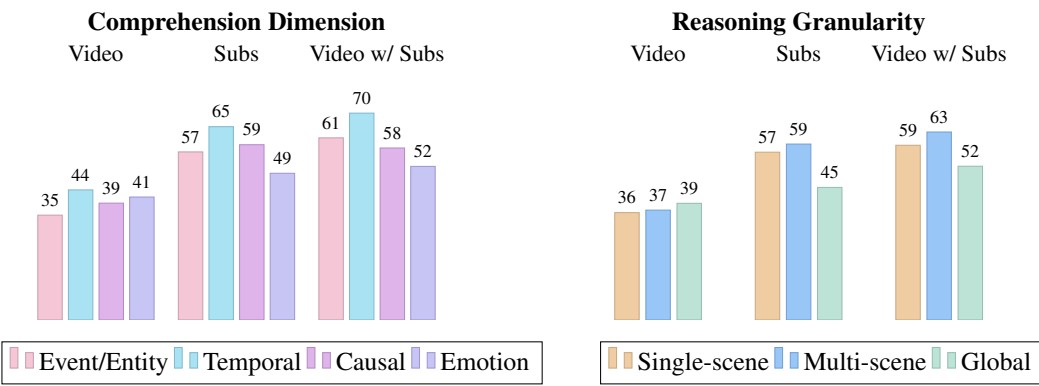

Figure 4: Pairwise accuracy for Gemini 2.5 Pro per comprehension dimension and reasoning granularity when varying the input modalities.

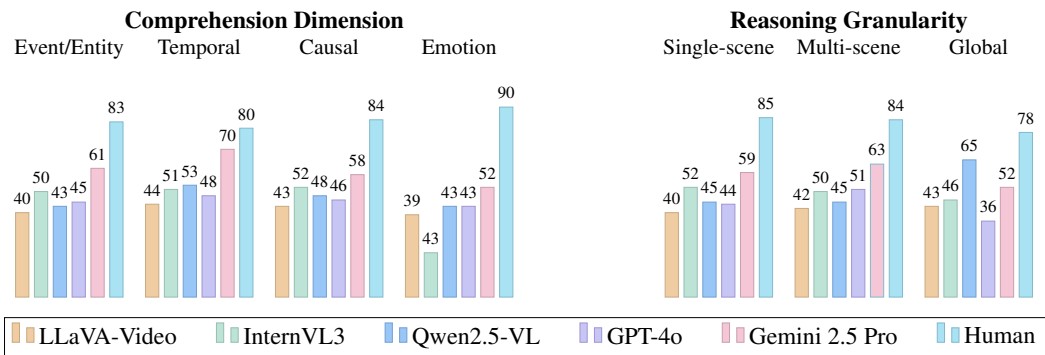

Figure 5: Pairwise accuracy for large-scale models with video and subtitles, and human baseline per comprehension dimension and reasoning granularity.

The results also highlight varying degrees of difficulty across the tasks, with emotion comprehension appearing to be a strong point for humans, while temporal perception is a strong point for models. Interestingly, the analysis on reasoning granularity reveals an interesting pattern between humans and models: as reasoning shifts from single-scene to multi-scene and eventually to global, model performance tends to oscillate across models, while human performance declines. Notably, Qwen2.5-VL shows improved accuracy on claims requiring global reasoning compared to the other granularities. This may suggest that global narrative information is more frequently represented in pretraining corpora (e.g., Wikipedia summaries of movies), whereas single-scene questions demand localized details that are less likely to be encountered in such sources. In contrast, humans may face increased cognitive load or memory limitations when reasoning across multiple scenes, which could explain the drop in performance in some cases.

## 5 RELATED WORK

**Vision and long context LLMs.** The field of VLMs has seen rapid progress, with models becoming increasingly effective at video-language understanding (Deitke et al., 2024; Bai et al., 2025; Zhu et al., 2025). Early methods focused on short clips and relied on complex spatio-temporal modules, such as Q-formers (Zhang et al., 2023a; Li et al., 2023a), or temporal pooling techniques (Maaz et al., 2024; Luo et al., 2023; Xu et al., 2024). While not new, projection layers (Li et al., 2023c; Liu et al., 2023; Li et al., 2023a; Liu et al., 2024) have gained popularity as a simpler and increasingly effective alternative for aligning video and language representations (Bai et al., 2025; Zhang et al., 2025; Zhu et al., 2025), largely driven by advancements in visual encoders (Radford et al., 2021; Tschannen et al., 2025). In the domain of long video understanding, current approaches primarily focus on compressing tokens (Li et al., 2023c; Zhang et al., 2025), merely extending the context window

(Abdin et al., 2024; Liu et al., 2025) or memory consolidation mechanisms (Balazevic et al., 2024; Song et al., 2023; 2024; Santos et al., 2025). A separate line of work first densely captions videos and then answers questions based on text only (Zhang et al., 2024; Wang et al., 2024c;e); we focus instead on benchmarking VLMs without costly captioning pipelines by introducing a benchmark that evaluates deep video understanding rather than simple memorization.

**Long video understanding benchmarks.**   Understanding long videos presents substantial challenges, requiring models to track complex temporal dependencies and retain narrative context over extended durations. While existing benchmarks have driven progress in temporal reasoning over short clips (Xiao et al., 2021; Wu et al., 2021) and in domain-specific settings such as instructional or egocentric videos (Yang et al., 2021; Mangalam et al., 2023; Qiu et al., 2024), most focus on content under three minutes or can be solved with a few keyframes (Yu et al., 2019; Zhang et al., 2023b). Benchmarks targeting longer content, such as (Mangalam et al., 2023; Rawal et al., 2024; Parmar et al., 2024; Wu et al., 2024; Hu et al., 2025b; Shafique et al., 2025), still fall short in average duration, scale, or annotation quality. Even those with longer videos (e.g., HourVideo (Chandrasegaran et al., 2024), InfiniBench (Ataallah et al., 2024)) often rely on synthetic questions and automated labels, and most use multiple-choice formats (e.g., Video-MME (Fu et al., 2024), LVBench (Wang et al., 2024b), Video-MMMU (Hu et al., 2025a)), which introduce biases and limit the assessment of genuine multimodal understanding. While (Huang et al., 2020) offers a dataset for long-form movie understanding, it provides only keyframes, which constrains the flexibility of evaluation. Similarly, SynopGround (Tan et al., 2024) and Timescope (Zohar et al., 2025) focus on long videos, but primarily target localized ("needle-in-a-haystack") retrieval rather than deep understanding. Neptune (Nagrani et al., 2025) pushes towards free-form answers and reasoning over long time horizons but remains limited to 15-minute videos; in the same vein, VideoAutoArena Luo et al. (2024) avoids multiple-choice evaluation by simulating users to rank long-form answers. Similarly, CG-Bench (Chen et al., 2024a) recognizes the limits of multiple-choice formats and evaluates models based on their ability to ground their answer to clues in the video. Critically, none of these datasets include claim pair tasks needed to assess a model's ability to integrate and create an intrinsic understanding across multi-hour content. Our benchmark's design—centered on long-form, manually annotated movie narratives and a binary claim evaluation protocol—offers a rigorous framework for diagnosing true narrative understanding in video-language models.

## 6  CONCLUSIONS

In this paper, we introduce $\mathbf{MF}^2$, a comprehensive multimodal benchmark designed to evaluate VLMs on deep narrative understanding in the context of long movie comprehension. Our benchmark adopts a binary evaluation protocol and covers a diverse range of claim categories, including emotion understanding, temporal perception, causal reasoning, and event/entity understanding. These claims span varying levels of granularity—single-scene, multi-scene, and global—requiring reasoning across entire films. All examples are annotated by humans to ensure high-quality and reliable labels. Our extensive evaluation of both open-weight and closed state-of-the-art models reveals a significant performance gap between models and humans, underscoring the challenges and importance of our benchmark. Commercial models such as Gemini 2.5 Pro outperform others, including GPT-4o and other open-weight variants, yet still fall short of human-level performance. We observe that incorporating transcripts significantly boosts model accuracy. Interestingly, Gemini 2.5 Pro decreases performance on questions requiring global reasoning, suggesting that our framework effectively targets the harder challenge of global narrative understanding, which current models continue to struggle with despite good overall capabilities. We hope MF$^2$ boosts future research and development aimed at improving the narrative reasoning capabilities of VLMs.

## 7  LIMITATIONS

Despite careful design and validation, our dataset is not free from imperfections. Minor issues such as typos may remain, and annotators—though shown one claim at a time—may have recalled earlier claims from the same pair, influencing later judgments. Models do not share this limitation, as they process claims independently. As future work, claims from each pair could be split into disjoint sets and rated by different annotators to better isolate such effects.

## 8 ETHICS STATEMENT

We adhered to established scientific and ethical standards in constructing and releasing $MF^2$. All source movies are released under the permissive Public Domain 1.0 license. Claims and annotations were created and validated exclusively by the authors; no external crowdworkers were employed. To encourage a plurality of perspectives in the annotation process, the annotation team consists of individuals from diverse demographic, institutional, and geographic backgrounds. Since $MF^2$ is derived entirely from fictional movies, it contains no personally identifiable information (PII) of real individuals. Nonetheless, some fictional content may reflect cultural stereotypes or outdated social norms. We caution researchers that models evaluated on $MF^2$ may inherit such biases, and we recommend appropriate safeguards when interpreting or deploying results. We advise users to employ $MF^2$ strictly within the scope of this work, namely as a benchmark for evaluating vision–language models on long movie understanding, and discourage its use beyond it.

## 9 REPRODUCIBILITY STATEMENT

We ensure reproducibility by releasing the full dataset and the codebase at https://anonymous.4open.science/r/MF2. The repository includes extended instructions to replicate all experimental settings. To facilitate long-term accessibility, and in accordance with the Public Domain 1.0 license, we additionally host a copy of the raw movie data. Detailed annotation protocols are provided in Appendix B, while Appendix C outlines additional experimental details. We encourage independent verification of our results and welcome contributions from the community to extend or stress-test $MF^2$ over time.

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

## A    METADATA FOR COLLECTED MOVIES

In Table 7, we provide detailed information on the 53 released movies, including their genre, original language, and duration.

## B    DETAILED GUIDELINES FOR DATA ANNOTATION AND HUMAN-EVAL

### B.1    DATA ANNOTATION GUIDELINES

In Figs. 6 and 7, we present the detailed guidelines provided to annotators during the data annotation process. These include instructions for constructing contrastive claim pairs, and labeling each pair with the appropriate reasoning granularity and comprehension dimensions. Furthermore, in Figs. 8 and 9, we include a subset of illustrative examples shown to annotators to guide their annotations of reasoning granularity and comprehension dimensions, respectively.

We note that among the comprehension dimensions annotators could assign to each claim pair, an "Other" category was included to account for cases that did not clearly align with any of the predefined dimensions. As this label was selected rarely (0.49% of the data), it is excluded from the figures presented in the main text.

### B.2    HUMAN EVALUATION GUIDELINES

In Figs. 10 and 11, we provide the full set of guidelines shared to participants during the human evaluation process, which consists of two stages: an initial stage in which evaluators respond without revisiting the movie, and an optional second stage that allows revisiting. While we only analyze the results from Stage 1—as our goal is to assess movie understanding based on memorable events without allowing participants to rewatch parts of the film—we include the complete instructions for both stages to offer full context. Additionally, we provide an illustration of the evaluation interface to clarify the evaluation setup.

## C    DETAILS ON EXPERIMENTAL SETUP

### C.1    PROMPT TEMPLATES

In Figs. 12 and 13 we present the direct and explanation prompt templates used for open-weight and closed models, respectively. The former requests only a True/False response, while the latter additionally asks for a brief justification before the final answer. We found that the direct prompt yielded better performance for open-weight models, while the explanation prompt proved more effective for closed models. When experimenting with different input modalities—such as adding the synopsis, subtitles, or movie title—we adapt the prompts accordingly.

### C.2    RESOURCES

Our infrastructure consists of a single machine equipped with 4 NVIDIA H100 GPUs (80GB each) and 12 Intel Xeon Gold 6348 CPUs (2.60GHz, 1TB RAM). All experiments were conducted on a single GPU, except for evaluations involving larger open-weight models (>70B parameters), where all 4 GPUs were used to accelerate inference.

## D    ADDITIONAL EXPERIMENTS

In Table 5 we provide the main results by including a greater coverage of models.

## E    ABLATION ON VISUAL STREAM USAGE

Given the moderate gains observed when models are provided with both video and subtitles compared to subtitles alone, we conduct an additional experiment with Qwen2.5VL-72B to assess whether

Table 5: Performance of both open-weight and closed models when evaluated on MF$^2$. We report both pairwise and standard accuracy, when models are assessed on video inputs w/ and w/o subtitles. Best-performing values among models are **bolded** and best for each specific group are underlined. Models with (*) fail to follow the instruction prompt.

| Method | #Params | #Frames | Pairwise Accuracy (%) | | Accuracy (%) | |
|---|---|---|---|---|---|---|
| | | | w/o subs | w/ subs | w/o subs | w/ subs |
| *Baselines* | | | | | | |
| Random | - | - | 25.0 | 25.0 | 50.0 | 50.0 |
| Human | - | - | - | **84.1** | - | **90.5** |
| *Closed Models* | | | | | | |
| GPT-4o | - | 50 | 18.8 | 46.8 | 55.2 | 71.4 |
| Claude 3.7 Sonnet | - | 100 | 3.80 | 44.6 | 51.4 | 71.5 |
| Gemini 2.5 Pro | - | 120 | **37.2** | **60.6** | **64.2** | **77.6** |
| *Open-weight Models* | | | | | | |
| Kangaroo* | 8B | - | - | - | - | - |
| LLaVA-Cinepile* | 7B | 8 | - | - | - | - |
| VideoLLaMA3 | 7B | 180 | 20.5 | 33.5 | 57.0 | 62.7 |
| Qwen2.5-VL | 7B | 180 | 24.6 | 32.8 | 56.7 | 62.0 |
| LLaVA-Video | 7B | 64 | 6.6 | 19.0 | 51.7 | 57.8 |
| LongVILA-R1 | 7B | 180 | 11.5 | 16.9 | 50.1 | 56.6 |
| InternVL3 | 8B | 64 | 10.9 | 36.9 | 53.1 | 64.6 |
| LLaVAOneVision1.5 | 8B | 32 | 4.9 | 26.4 | 51.0 | 60.9 |
| Gemma3 | 27B | 64 | 31.5 | 42.9 | 61.2 | 68.1 |
| Qwen3VL-IT | 30B | 180 | 19.2 | 42.3 | 56.9 | 68.6 |
| Ovis2 | 34B | 10 | 18.8 | 45.6 | 53.3 | 69.5 |
| InternVL3.5 | 38B | 64 | 26.6 | 46.2 | 60.0 | 70.3 |
| Qwen2.5-VL | 72B | 180 | 29.7 | 45.9 | 58.8 | 70.4 |
| LLaVA-Video | 72B | 64 | 15.6 | 41.8 | 54.6 | 69.1 |
| InternVL3 | 78B | 64 | 22.1 | 51.3 | 58.0 | 72.7 |

Table 6: Performance of Qwen2.5VL-72B under shuffled and non-shuffled visual inputs.

| Video Only | | | | |
|---|---|---|---|---|
| **Setting** | **Pairwise Acc.** | **Avg Acc.** | **Acc. (Facts)** | **Acc. (Fibs)** |
| Original frames | 29.7 | 58.8 | 45.4 | 72.1 |
| Shuffled frames | 4.8 | 51.8 | 4.9 | 98.7 |
| **Video + Subtitles** | | | | |
| **Setting** | **Pairwise Acc.** | **Avg Acc.** | **Acc. (Facts)** | **Acc. (Fibs)** |
| Original frames | 45.9 | 70.4 | 55.7 | 85.1 |
| Shuffled frames | 40.3 | 66.2 | 46.1 | 86.3 |

the visual stream is being used. To do so, we shuffle the sampled video frames and compare performance against the non-shuffled setting. We run this experiment for both the video-only and the video-plus-subtitles setups. We report pairwise accuracy, average accuracy, fact (true-claim) accuracy, and fib (false-claim) accuracy. Table 6 presents the results. The substantial drop in pairwise accuracy and fact accuracy under frame shuffling indicates that the model does indeed use visual cues. This observation is further supported by the examples in Appendix F. Gemini's explanations for its predictions explicitly reference the visual cues, confirming that the model makes use of the visual stream.

## F  ILLUSTRATIVE EXAMPLES

In this part, we include a small set of examples (Tables 8, 9, 10, 11, 12, 13, 14, 15, 16, 17) showing when visual or text cues are needed to resolve the claims and how Gemini 2.5 Pro (best performing model) succeeds or fails on them, along with its corresponding explanations.

## AI ASSISTANCE

We would like to note that large language models (ChatGPT) were used to assist in drafting and polishing the writing of this work.

Table 7: Details of collected movies.

| Movie (Year) | Genre (IMDB) | Language | Duration (mins) |
|---|---|---|---|
| The Last Chance (1945) | Drama, War | en, it | 93.84 |
| They Made Me a Criminal (1939) | Boxing, Film Noir, Crime, Drama, Sport | en | 91.21 |
| Tokyo After Dark (1959) | Drama | en | 81.23 |
| The Sadist (1963) | Horror, Thriller | en | 91.63 |
| Suddenly (1954) | Film Noir, Psychological Thriller, Crime, Drama, Thriller | en | 76.71 |
| Sabotage (Hitchcock) (1936) | Psychological Thriller, Spy, Crime, Thriller | en | 75.92 |
| Murder By Contract (1958) | Film Noir, Crime, Drama, Thriller | en | 80.45 |
| Pushover (1954) | Film Noir, Crime, Drama, Thriller | en | 87.77 |
| Go for Broke (1951) | Drama, History, War | en | 90.85 |
| Meet John Doe (1941) | Political Drama, Satire, Comedy, Drama, Romance | en | 122.87 |

| | | | |
|---|---|---|---|
| Scarlet Street (1945) | Film Noir, Tragedy, Crime, Drama, Thriller | en | 102.39 |
| Little Lord Fauntleroy (1936) | Period Drama, Drama, Family | en | 100.72 |
| Deadline - U.S.A. (1952) | Film Noir, Crime, Drama | en | 87.06 |
| My Favorite Brunette (1947) | Hard-boiled Detective, Comedy, Crime, Mystery, Romance, Thriller | en | 87.34 |
| Woman in the Moon (1929) | Adventure, Comedy, Drama, Romance, Sci-Fi | de | 168.73 |
| Lonely Wives (1931) | Comedy, Romance | en | 85.35 |
| Nothing Sacred (1937) | Satire, Screwball Comedy, Comedy, Drama, Fantasy, Romance | en | 73.57 |
| Fingerman (1955) | Film Noir, Crime, Drama, Thriller | en | 82.06 |
| Borderline (1950) | Film Noir, Crime, Drama, Thriller | en | 88.16 |
| Babes in Toyland (1934) | Screwball Comedy, Slapstick, Comedy, Family, Fantasy, Musical | en | 77.26 |
| The Man From Utah (1934) | Drama, Western | en | 51.49 |
| The Man With The Golden Arm (1955) | Drug Crime, Psychological Drama, Crime, Drama, Romance | en | 119.07 |
| A Star Is Born (1937) | Tragic Romance, Drama, Romance | en | 110.98 |
| Africa Screams (1949) | Farce, Action, Adventure, Comedy | en | 79.13 |
| Dementia 13 (1963) | Slasher Horror, Horror, Thriller | en | 74.94 |
| Fear and Desire (1952) | Drama, Thriller, War | en | 70.19 |
| The Little Princess (1939) | Costume Drama, Comedy, Drama, Family, Musical | en | 92.77 |
| Father's Little Dividend (1951) | Comedy, Drama, Romance | en | 81.74 |
| Kansas City Confidential (1952) | Conspiracy Thriller, Film Noir, Heist, Crime, Drama, Thriller | en | 99.27 |
| Of Human Bondage (1934) | Dark Romance, Film Noir, Medical Drama, Tragedy, Tragic Romance, Drama, Romance | en | 82.77 |
| Half Shot at Sunrise (1930) | Comedy, Musical | en, fr | 78.04 |
| Bowery at Midnight (1942) | B-Horror, Crime, Horror, Thriller | en | 62.05 |
| The Emperor Jones (1933) | Drama, Music | en | 76.29 |

| | | | |
|---|---|---|---|
| The Deadly Companions (1961) | Adventure, Drama, Western | en | 93.62 |
| The Red House (1947) | Film Noir, Drama, Mystery, Thriller | en | 100.39 |
| Trapped (1949) | Film Noir, Crime, Drama, Thriller | en | 79.4 |
| City of Fear (1959) | Crime, Drama, Thriller | en | 75.18 |
| Kid Monk Baroni (1952) | Action, Drama, Sport | en | 79.56 |
| Tight Spot (1955) | Film Noir, Crime, Drama, Thriller | en | 95.99 |
| Captain Kidd (1945) | Costume Drama, Swashbuckler, Adventure, Biography, Drama, History | en | 87.53 |
| The Front Page (1931) | Dark Comedy, Satire, Screwball Comedy, Comedy, Crime, Drama, Mystery, Romance | en | 101.14 |
| The Hitch-Hiker (1953) | Film Noir, Crime, Drama, Thriller | en | 70.8 |
| Obsession (1949) | Film Noir, Psychological Thriller, Crime, Thriller | en | 92.39 |
| Thunderbolt (1929) | Film Noir, Crime, Drama, Music, Romance | en | 91.27 |
| Cyrano de Bergerac (1950) | Swashbuckler, Adventure, Drama, Romance | en | 112.87 |
| Scandal Sheet (1952) | Film Noir, Crime, Drama, Romance, Thriller | en | 81.75 |
| Ladies in Retirement (1941) | Film Noir, Crime, Drama | en | 92.31 |
| Detour (1945) | Film Noir, Crime, Drama | en | 69.09 |
| The Crooked Way (1949) | Film Noir, Crime, Drama, Thriller | en | 85.95 |
| A Bucket of Blood (1959) | Comedy, Crime, Horror | en | 65.84 |
| Love Affair (1939) | Holiday Romance, Comedy, Drama, Romance | en | 89.62 |
| The Jackie Robinson Story (1950) | Biography, Drama, Sport | en | 76.82 |
| The Last Time I Saw Paris (1954) | Tragedy, Tragic Romance, Drama, Romance | en | 116.02 |

1188
1189
1190
1191
1192
1193
1194
1195
1196
1197
1198
1199
1200
1201
1202
1203
1204
1205
1206
1207
1208
1209
1210
1211
1212
1213
1214
1215
1216
1217
1218
1219
1220
1221
1222
1223
1224
1225
1226
1227
1228
1229
1230
1231
1232
1233
1234
1235
1236
1237
1238
1239
1240
1241

## Guidelines for Data Annotation (Part 1)

We are conducting a research study on long movie understanding as part of a broader effort to explore how well viewers comprehend and recall complex narratives. Your task is to create claims that test a viewer's comprehension of a movie after watching it. These claims will be used in a human evaluation study to assess how well participants understand and recall key events from the movie. We appreciate your participation in this data collection process.

**General Task Instructions**   Select a movie from the current "Pool" of movies (the "Pool" can be found in <LINK>). Make sure this movie is not selected by another annotator.

- Watch the entire movie carefully.
- We highly recommend reading the example claims provided to gain a better understanding of the task you need to fulfil.
- Start writing down your claims following the template available in <LINK> (you will find two tabs available: the "Examples" tab contains claim examples, and the "Annotations Template" tab is the template you should follow). Please create another sheet with your claims–do not directly use the current template–and send it to us once it is completed.

**Annotation Process**

**1. Writing Claims**   You are asked to create pairs of contrastive claims, where one claim is true (fact) and the counterfactual version is false (fib). The two claims should differ by minimal edits, meaning they should be as similar as possible while maintaining contrast. Each claim should differ in a subtle but meaningful way, challenging comprehension without being overly obvious.
**Example:**
Fact: The first bomb exploded in the bus.
Fib: The first bomb exploded in the aquarium.
Why this works: The counterfactual claim is created with minimal edits, maintaining contrast while testing the understanding of a key event.

**2. Select Claim Granularity**   For each pair of claims you constructed, indicate whether answering them correctly requires reasoning based on a single scene, multiple scenes, or globally within the movie.
**Definition of scene:**
A scene in film refers to a complete unit of storytelling, usually consisting of a sequence of events and dialogue taking place in a specific location and time. It often involves one or more characters and is usually shot in one continuous take or consisting of a sequence of shots.
**Reasoning Granularity Labels:**

- **Single-scene:** Claims that are answerable using information from a single scene.
- **Multi-scene:** Claims falling into this granularity require information/evidence from multiple distinct scenes, but not from the whole film. In this case, details are usually spread out between the multiple scenes. The supporting information/evidence is distributed, but explicit and locatable (timestamps/scenes can be clearly identified and referenced)
- **Global:** Claims falling into this granularity require a holistic understanding of the movie narrative. They cannot be easily tied to specific scenes or timestamps, and need to infer or accumulate information/evidence that emerges across the entire narrative (timestamps/scenes can not be clearly identified and referenced).

*Note:* Reasoning granularity labels should be selected based on the fact (true claim). Check the examples provided in the "Examples for Reasoning Granularity" part.

Figure 6: Guidelines provided for the data annotation procedure (Part 1).

---

**Guidelines for Data Annotation (Part 2)**

**3. Claim Categorization**    Identify the comprehension dimensions the constructed pair of claims examines. Sometimes more than one dimension is examined, so we allow for multiple labels.

**Comprehension Dimension Labels:**

- **Event/Entity Understanding:** it refers to claims that require the identification of key entities (such as people, places, or objects) and understanding of actions or events involving those entities throughout the narrative. Understanding these claims involves tracking the presence and role of entities across scenes, extracting relationships among them, observing and interpreting their actions, and linking them to relevant events in the narrative.

- **Temporal Perception:** temporal perception refers to claims that require understanding of the timeline of events. It involves reasoning about the order in which events or actions occur—e.g., determining whether an event/action takes place before, after or at the same time as another—and may also require counting the number of specific actions or events. Unlike tasks focused on localizing a specific action in time, temporal perception emphasizes comprehension of broader temporal relationships within the evolving storyline.

- **Emotion Understanding:** emotional understanding refers to claims that involve recognizing and interpreting the emotional development of characters throughout the narrative.

- **Causal Reasoning:** causal reasoning refers to claims that require identifying cause-and-effect relationships between events or actions, where the relationship may be either direct or implicit.

- **Other:** If none of the above fit, select "Other" and suggest a new category.

*Note:* The categorization is based on both claims (fact and fib). Check the examples provided in the "Examples for Comprehension Dimensions" part.

**Important Points To Consider**

- **Ensure claims assess the viewer's understanding of the movie**. To put it simply, claims should refer to **significant moments** in the movie, **avoiding trivial details or Needle in a Haystack (NIAH)-style claims**, such as: "The detective wears a red T-shirt" (if this detail is not important in the movie).

- **Claims must be clear and unambiguous in isolation**, meaning they should be understandable without requiring additional context but should still require reasoning based on the movie. **Each claim should be self-contained and make sense independently**, without referencing its counterfactual version. Also, **avoid highly subjective or interpretive claims**. Each claim should still have a definitive answer based on the movie's content.

- **Avoid providing unnecessary contextual details**. For example, do not use phrases like "in the beginning of the movie, . . .", "in the final scene, . . ." unless such information is essential to understanding the claim.

- Ensure that claims **span the entire movie** rather than focus on isolated scenes.

- Once you finish the annotation process, please **go through your claims and confirm that they are in line with the points raised above** (these points are important to be covered to ensure good quality of annotations).

Figure 7: Guidelines provided for the data annotation procedure (Part 2).

**Guidelines for Data Annotation (Part 3)**

**Examples for Reasoning Granularity**    In this part, we provide examples to illustrate how to assign reasoning granularity labels.

**Example 1:**
*Fact:* According to the Hattley, the individual shown in the photograph (Marakelli) worked with Constain.
*Fib:* According to Hattley, the individual shown in the photograph (Marakelli) had no connection or working relationship with Constain.
*Reasoning Granularity:* Single-scene.
*Justification:* This event is categorized as single-scene because it takes place within one specific scene: Hattley shows the photograph to Conley, they are having a discussion and it is implied that Marakelli worked with Constain in the mafia.

**Example 2:**
*Fact:* Hattley appeared visibly bothered with the discussion he had in his office with Constain's attorney.
*Fib:* Hattley appeared pleased with the discussion he had in his office with Constain's attorney.
*Reasoning Granularity:* Single-scene.
*Justification:* That is again a single scene event. Constain's attorney enters the office and they are having a discussion. After a while, Hattley kicks him out.

**Example 3:**
*Fact:* Miss Conley received a dress as a personal gift from the policeman.
*Fib:* Miss Conley received a dress as a gift from the government, delivered by the policeman.
*Reasoning Granularity:* Multi-scene.
*Justification:* That is a multi-scene event, that we need to ground on 2 independent scenes to answer the question correctly. In the first scene Miss Conley receives a gift from the policeman, who says that the gift is from the government. After a while (some scenes are interleaved), she understands that the policeman bought the gift for her and not the government. So to answer correctly, we need to ground on these 2 specific scenes.

**Example 4:**
*Fact:* Conley's statement about her occupation, describing herself as a "gang buster," implicitly refers to Constain.
*Fib:* Conley's statement about her occupation, describing herself as a "gang buster," implicitly refers to Pete Tinelli.
*Reasoning Granularity:* Global
*Justification:* There is a single scene in the end of a movie during which Conley characterises herself as a "gang buster". Although it is a single scene, it is impossible to understand solely by this scene why she said it and to whom she is referring to. We need to watch a big part of the movie (if not all of it) to understand that refers to Constain.

Figure 8: Guidelines provided for the data annotation procedure (Part 3). This part of the guidelines provides examples given to annotators to illustrate how to assign reasoning granularity labels. While more examples were shared during the annotation process, we include a selection here for illustrative purposes.

1350
1351
1352
1353
1354
1355
1356
1357
1358
1359
1360
1361
1362
1363
1364
1365
1366
1367
1368
1369
1370
1371
1372
1373
1374
1375
1376
1377
1378
1379
1380
1381
1382
1383
1384
1385
1386
1387
1388
1389
1390
1391
1392
1393
1394
1395
1396
1397
1398
1399
1400
1401
1402
1403

**Guidelines for Data Annotation (Part 4)**

**Examples for Comprehension Dimensions**   In this part we provide examples to illustrate how to assign comprehension dimension labels.

**Example 1:**
*Fact:* At Jim's bar, the Connel keeps drinking as he talks to the fake John Doe, expressing his frustration and concern.
*Fib:* At Jim's bar, the Connel keeps drinking as he talks to the fake John Doe, expressing hope and happiness.
*Comprehension Dimension:* emotion understanding
*Justification:* We need to understand what emotion Connel expressed, to answer the pair of claims correctly.

**Example 2:**
*Fact:* Conley's statement about her occupation describing herself as a "gang buster", implicitly refers to Constain.
*Fib:* Conley's statement about her occupation describing herself as a "gang buster", implicitly refers to Pete Tinelli.
*Comprehension Dimension:* entity/event understanding
*Justification:* We need to understand to whom the expression "gang buster" refers to. So, the comprehension dimension is entity understanding.

**Example 3:**
*Fact:* Hallet brought Conley's sister to the hotel with the intent to make Conley testify in the trial.
*Fib:* Hallet brought Conley's sister to the hotel with the intent to make her feel safe.
*Comprehension Dimension:* causal reasoning
*Justification:* Here we need to understand why Hallet brought Conley's sister to the hotel. So it examines a causal-and-effect relationship.

**Example 4:**
*Fact:* Conley decided to testify only after Wiloughby's death.
*Fib:* Conley had already decided to testify before Wiloughby's death.
*Comprehension Dimension:* temporal perception
*Justification:* that pair examines the temporal dimension (if the decision was taken before or after Wiloughby's death).

Figure 9: Guidelines provided for the data annotation procedure (Part 4). This part of the guidelines provides examples given to annotators to illustrate how to assign comprehension dimension labels. While more examples were shared during the annotation process, we include a selection here for illustrative purposes.

**Guidelines for Human Evaluation (Part 1)**

This evaluation study aims to assess how well people comprehend and recall key events from a movie. You will watch a movie and then evaluate a series of claims about its content. Your goal is to determine whether each claim is True or False, based solely on what was shown in the movie. We appreciate your participation in this study.

**Task Instructions**

- Assign to yourself the movies you want to watch and do the test (we expect 2 movies per person). Please add your name to the Human-Eval column, on this LINK.

- Visit the platform for evaluation LINK.

- Provide your email to receive access to the movie (it will be used as your unique identifier).

- Once you submit your email, you should carefully select from the drop-down list the corresponding movie you assigned yourself and proceed with the evaluation. You will be shown with the movie link. Please open it in a new tab.

The test is divided in **2 stages**: The **first stage** is **mandatory** and should be completed by everyone (*during this stage you are not allowed to go back to the movie while answering the questions*). The **second stage** is **optional** (*during this stage you are allowed to go back to the movie while answering the questions*).

**Stage 1:**

1. **Watch the entire movie carefully before proceeding to the evaluation**. Pay attention to details and context in the movie, as some claims may be subtle or require careful reasoning.

2. After watching, it's time to proceed to Stage 1. **Please do not go back to the movie until Stage 1 of the test is completed.** Press the "Start Classifying Claims" button, and you will be shown with **one claim at a time**. For each claim shown, you need to do the following:

   - **Classify the claim as True/False** (you should always answer truthfully, without aiming to maximise you score).
   - Mark your **confidence** about your answer. This is helpful for stage 2, where you will have the opportunity to revise your claims (by looking back at the movie).
   - Leave a comment if any of the following applies: If a claim is **ambiguous, unclear, open to interpretation, has a bad phrasing or typos, you may leave an optional comment explaining your concerns**. You can also comment on the claim in case it is **needle-in-a-haystack style** and you think it is too detailed and doesn't test the understanding of the movie.
   - Once you answered, click "Save" to submit your response and move on to the next claim.

**Important details:** Once you submit an answer, you cannot go back and change it. At this stage, you are **strictly prohibited from searching back in the movie, rewinding, or rewatching scenes while answering the claims**. Your responses **should be based on your memory** and **understanding**. You must **NOT use any AI tools or external sources to verify or generate answers**. The goal of this study is to assess human understanding of long movies, not automated retrieval or AI-assisted responses. Also you are not allowed to take any paper notes, while watching the movie.

Figure 10: Guidelines provided for human evaluation (Part 1).

**Guidelines for Human Evaluation (Part 2)**

**Stage 2:**
Once you complete Stage 1, you will see a message asking you if you want to proceed to Stage 2 (Stage 2 is optional).

During Stage 2, you will be shown again with the choices you selected during Stage 1, but now **you can revise your answers by looking back to the movie** (you can reuse the movie link we provided you). You will be shown for each claim with the choices you did in Stage 1. You are free to change them and proceed to the next claims. Don't worry your answers will not be overwritten. Once you finish with Stage 2, you will be shown with a confirmation message.

If you have any questions or encounter any technical issues, please report them to our team! Thank you for your time and effort!

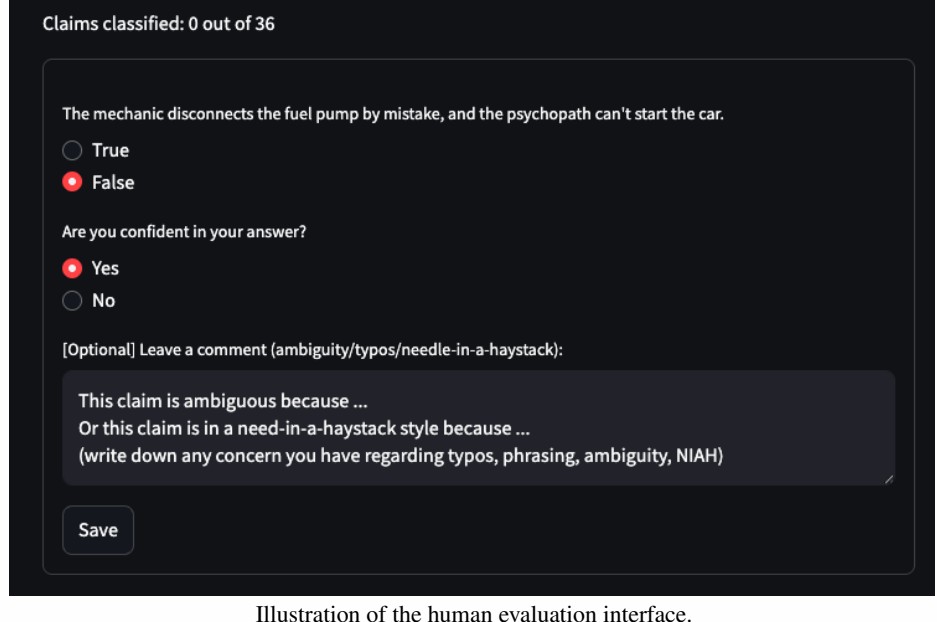

Illustration of the human evaluation interface.

Figure 11: Guidelines provided for human evaluation (Part 2).

Direct Prompt Template

**System:** You are a helpful AI assistant. Your task is to carefully analyze the provided content and determine whether statements made about it are true or false based on the available information.

**User:** You are provided with a movie and a statement. Your task is to carefully watch the movie and then determine whether the statement is true or false.
Answer TRUE if the statement is true in its entirety based on the movie.
Answer FALSE if any part of the statement is false based on the movie.

**Statement: {claim}**
Based on the movie, is the above statement TRUE or FALSE?
Provide only your final answer.

Figure 12: Direct prompt template used for **open-weight** models.

Explanation Prompt Template

**System:** You are a helpful AI assistant. Your task is to carefully analyze the provided content and determine whether statements made about it are true or false based on the available information.

**User:** You are provided with a movie and a statement. Your task is to carefully watch the movie and then determine whether the statement is true or false.
Answer TRUE if the statement is true in its entirety based on the movie.
Answer FALSE if any part of the statement is false based on the movie.

**Statement: {claim}**
Based on the movie, is the above statement TRUE or FALSE?
First provide an explanation of your decision-making process in at most one paragraph, and then provide your final answer.

Figure 13: Explanation prompt template used for **closed** models.

Table 8: **Illustrative Example 1 from the movie "Suddenly"**. Gemini fails to predict both claims correctly in every modality setting.

---

**Example 1 (Part 1) - Movie:** *"Suddenly"*

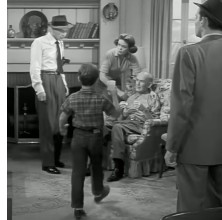 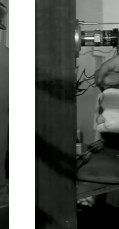 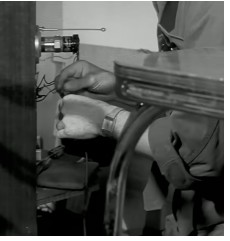 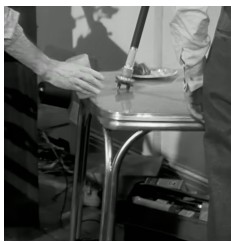 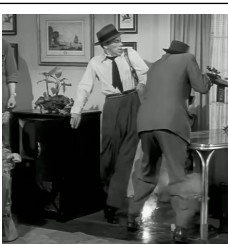

Pidge brings the water      Setting up the circuit      Pop spills the water      The electrocution

**Fact** (✓): The grandfather, Peter Benson, intentionally spilled the water on the floor.
**Fib** (✗): The grandfather, Peter Benson, accidentally spilled the water on the floor.

**Granularity**: Multi-scene

**Context (Spoiler alert !)**: Early in the film, the TV breaks. Later, when the criminals have already broken into the house, pop (the grandfather) orchestrates the setup to save everyone (without explicitly stating it). He feigns a heart attack and asks for water, Pidge brings it. The TV technician pretends to repair the broken TV as part of a sabotage plan, while the circuit. When the moment comes, Pop *intentionally* spills the water on the floor (placing it on the edge of the table). Shortly after, the electrocution occurs when one of the criminals steps on the spilled water. *Both visual and textual information are needed to understand that Pop indeed spilled the water and what was his intention. Gemini fails to predict both claims correctly in every modality setting.*

**Gemini's Predictions:**
*Subs only setting*

**Prediction on Fact**: The statement is "The grandfather, Peter Benson, intentionally spilled the water on the floor." To determine if this is true, we need explicit evidence in the transcript: 1. That Peter Benson spilled water; 2. That he did so intentionally. Transcript evidence: Peter Benson (Pop) feigns a heart attack. He asks Pidge: "Pidge, put down that gun and get some water for your grandfather." Pidge presumably brings the water. Later, just before the climax, the line "Oh, butterfingers!" appears. This implies something was dropped or spilled, likely the water. However, the transcript does not explicitly state who exclaimed "Oh, butterfingers!" or who spilled the water. Since the transcript does not explicitly confirm that Peter Benson spilled the water, the statement cannot be verified. **FALSE** *(Ground-truth: TRUE)*

**Prediction on Fib**: The transcript mentions Pidge being asked to get water for his grandfather, Peter Benson (Pop), when Pop feigns a heart attack [..] However, the transcript does not state that Peter Benson subsequently spilled this water, or any water, on the floor. [...] it is not explicitly stated who Jud is referring to or that the fumbling involved spilling water. The context is the electrocution and shooting, not an earlier incident of spilling water. There is no direct statement or clear implication in the transcript that Peter Benson accidentally spilled water on the floor. **FALSE** *(Ground-truth: FALSE)*

---

Table 9: **Illustrative Example 1 from the movie "Suddenly"**. Gemini fails to predict both claims correctly in every modality setting.

---

**Example 1 (Part 2) - Movie:** *"Suddely"*

---

*Video only setting*

**Prediction on Fact**: The grandfather, Peter Benson, is seen at 2:17 reaching for a glass of water. He then appears to deliberately drop the glass, causing the water to spill on the floor. This creates a distraction, allowing the woman to crawl towards a gun. The context of the scene, where the family is being held hostage and looking for an opportunity to fight back, combined with the grandfather's actions and the subsequent events, strongly suggests the spill was intentional. **TRUE** *(Ground-truth: TRUE)*

**Prediction on Fib**: The video introduces an older character who is later identified as Peter Benson (around 0:46-0:47). Much later in the video, during a tense hostage situation, Peter Benson is seen sitting in an armchair holding a glass of water. At timestamp 2:18, he is startled and gets up abruptly, causing the water in his glass to spill onto the floor. The context of the scene, where he reacts to the unfolding events, suggests the spill was accidental due to his sudden movement or reaction. **TRUE** *(Ground-truth: FALSE)*

*Video w/ subs setting*

**Prediction on Fact**: The grandfather, Peter Benson, is seen getting up with a glass of water around timestamp 55:20. As he walks, he stumbles and drops the glass, spilling the water on the floor near the table leg and electrical wiring. Immediately after, he clutches his chest and feigns a heart attack. The transcript reflects this with dialogue about his pain. There is no visual or textual evidence to suggest the spill was intentional; rather, it appears accidental, possibly due to his feigned or real physical distress, and he then uses the moment to further distract the captors by exaggerating his heart condition.**TRUE** *(Ground-truth: TRUE)*

**Prediction on Fib**: The grandfather, Peter Benson, takes a glass of water and walks towards the table where the assassin has set up the rifle. He places the glass on the edge of the metal table (around 55:23). As he moves to talk to Jud, who is working on the TV, the glass tips over and spills the water on the floor (around 55:24). The spilling appears to be an unintentional consequence of his actions and the precarious placement of the glass. Therefore, it was an accident.**TRUE** *(Ground-truth: FALSE)*

---

Table 10: **Illustrative Example 2 from "Sabotage (Hitchcock)"**. Gemini predicts the pair of claims incorrectly in every modality setting.

---

**Example 2 (Part 1) - Movie:** *"Sabotage (Hitchcock)"*

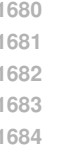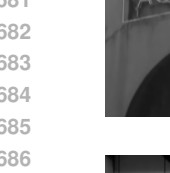 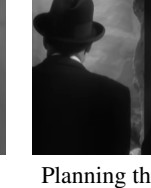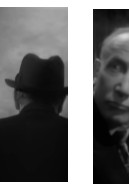 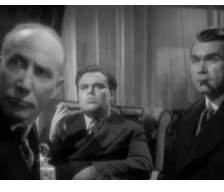 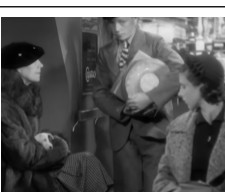

| Aquarium | Planning the sabotage | Meet in cinema | Kid enters the bus |

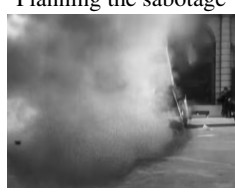 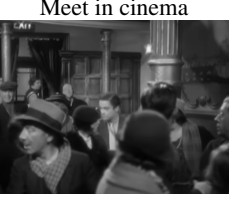 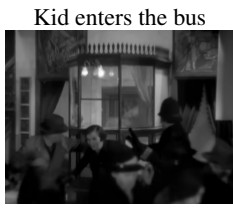

| The time arrives | Bus explosion | People in the cinema | Explosion in cinema |

**Fact** (✓): The second bomb exploded in the cinema.
**Fib** (✗): There was a second bomb about to explode, but it didn't.

**Granularity**: Global
**Context** (Spoiler alert !): Mr. Verloc and the terrorist meet at the aqaurium, where the terrorist first mentions the plan to explode a bomb (he mentions "a parcel of fireworks" and asks Mr. Verloc to meet his friend the bombmaker). The terrorist meet again in the cinema, to discuss about the sabotage. Later in the movie the explosion happens inside the bus. While the second explosion happens inside the cinema. *Gemini fails to predict both claims correctly in every modality setting.*

**Gemini's Predictions:**

*Subs only setting*

**Prediction on Fact**: To determine if the statement is true or false, we need to identify the sequence of bomb-related incidents in the transcript. 1. **First Incident (Sabotage at the Cinema):** The transcript begins with a power outage at the cinema, described as "Sand. Sabotage. Wrecking. Deliberate." This is the first act of sabotage or a device causing disruption that Verloc is connected with. [...] This incident occurred at the cinema. 2. **Second Incident (Stevie's Bomb on the Bus):** Mr. Verloc is then tasked with a more serious act: to leave a "small parcel... a parcel of fireworks" (a bomb) in Piccadilly Circus. His young brother-in-law, Stevie, is sent to deliver this parcel. This bomb explodes prematurely. [...] This is the second distinct sabotage/bomb event in the plot, and this bomb exploded on a bus, not in the cinema. 3. **Third Incident (Bomb Explosion at the Cinema):** Towards the end of the transcript, the old man who makes the bombs is cornered in the cinema. [...] This bomb then explodes: [...] This explosion occurs at the cinema. Based on this chronological sequence of events: * The first destructive "bomb-like" event was the sabotage at the cinema (power outage). * The second bomb (Stevie's parcel) exploded on a bus in the West End. * The third bomb (detonated by the old man) exploded in the cinema. [...] Therefore, the statement is false. **FALSE** *(Ground-truth: TRUE)*

**Prediction on Fib**: The transcript describes two main bomb-related incidents. The first is the explosion that kills Stevie. The second occurs at the cinema where the old man (the bomb maker) threatens to "blow the place sky high." The transcript later confirms an explosion there: "The roof's gone. The whole of the back of the place - clean blown off." This indicates the second bomb *did* explode. Therefore, the statement that a second bomb was "about to explode, but it didn't" is false because it did explode. **FALSE** *(Ground-truth: FALSE)*

---

Table 11: **Illustrative Example 2 from "Sabotage (Hitchcock)"**. Gemini predicts the pair of claims incorrectly in every modality setting.

---

**Example 2 (Part 2) - Movie:** *"Sabotage"*

---

*Video-only setting*

**Prediction on Fact**: The video shows one clear explosion involving a bus on a street around the 1:46 mark. Following this, there are various scenes, including one where people are watching a cartoon in a cinema. However, there is no depiction or indication of a second bomb exploding, nor is there any explosion shown occurring inside the cinema. **FALSE** *(Ground-truth: TRUE)*

**Prediction on Fib**: The video montage shows scenes from Alfred Hitchcock's "Sabotage." The primary bomb plot shown results in an explosion on a bus around 1:46. Later in the montage, around 2:17-2:19, the Professor (the bomb maker) is seen suspiciously entering the cinema building. This implies he might be carrying a second bomb and intending to cause another explosion. However, the video montage ends shortly after this, at 2:23, without showing what happens to the Professor or if there was indeed a second bomb that was "about to explode." Crucially, the video does \*not\* show this second bomb failing to explode or being prevented from exploding. Since the video does not provide any information about the outcome of this potential second bomb (neither showing it explode nor showing it \*not\* explode), the assertion "but it didn't" is not supported by the available video content. Therefore, the statement cannot be confirmed as true based on the provided video. **FALSE** *(Ground-truth: FALSE)*

*Video w/ subs setting*

**Prediction on Fact**: The first act of sabotage was the power station blackout. The second planned act of sabotage, referred to as "a parcel of fireworks", was to be left in the cloakroom at Piccadilly Circus underground station [...] This parcel, carried by Stevie on a bus, explodes on the bus at 1:45. A newspaper headline later confirms "WEST END BOMB SENSATION. Fully-laden bus was blown up in Piccadilly this afternoon". [...] An explosion then occurs at the cinema, as described by an officer: "The roof's gone. The whole of the back of the place - clean blown off". This is a separate, third bomb event. Therefore, the second bomb (the one Stevie was carrying) exploded on the bus, not in the cinema. The explosion in the cinema was a distinct event caused by a different bomb wielded by the bird shop owner. **FALSE** *(Ground-truth: TRUE)*

**Prediction on Fib**: The statement claims: "There was a second bomb about to explode, but it didn't." Let's analyze the bombs in the film: 1. The first act of sabotage is the power station blackout. While not explicitly shown as a bomb explosion, it is an act of "wilful destruction." 2. The second major plot device is the bomb intended for Piccadilly Circus on Lord Mayor's Show day. Verloc gives this bomb to Stevie. This bomb explodes prematurely on the bus, killing Stevie and many others [...]. So this bomb \*did\* explode. 3. The third instance involving a bomb is at the cinema, where the Professor (the bomb maker) is cornered by the police. [..] This bomb \*does\* explode [...]. In all instances where a bomb is primed or ready for its purpose within the narrative, it either explodes (bus bomb, Professor's bomb in the cinema) or fulfills its sabotage (power station). There is no scene in the film where a bomb is "about to explode" and then "didn't." The bombs that are set up to go off, do go off. Therefore, the statement is false. **FALSE** *(Ground-truth: FALSE)*

---

Table 12: **Illustrative Example 3 from "Scandal Sheet"**. The pair of claims relies primarily on visual information. Gemini predicts them incorrectly when given only a single modality, but when provided with both the transcript and the video, it correctly identifies the pair of claims.

---

**Example 3 (Part 1) - Movie: *"Scandal Sheet"***

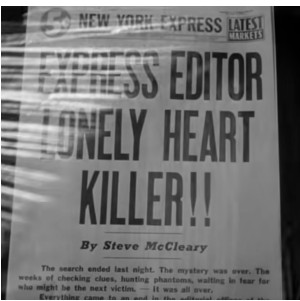 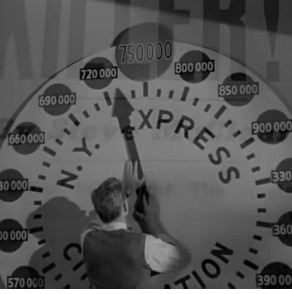 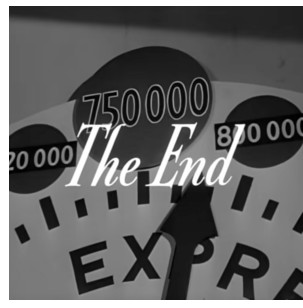

News about the killer          Sales increasing..          Sales increasing..

**Fact (✓)**: The sales of the newspaper increased after it was revealed that the editor was the killer.
**Fib (✗)**: The sales of the newspaper decreased after it was revealed that the editor was the killer.

**Granularity**: Single-scene

**Context (Spoiler alert !)**: In the film's climax, Steve McCleary discovers that his editor-in-chief, Mark Chapman, is the "Lonely Hearts Killer." After Chapman is exposed and confronted, McCleary—now acting as editor—publishes the full story, revealing Chapman's true identity and crimes. Following the revelation, the newspaper's sales surge dramatically. *This pair of claims relies primarily on visual information. Gemini predicts them incorrectly when given only a single modality, but when provided with both the transcript and the video, it correctly identifies the pair of claims.*

**Gemini's Predictions**:
*Subs only setting*

**Prediction on Fact**: The transcript ends with Mark Chapman, the editor, being apprehended as the killer. He tells the reporter McCleary to "Write it up big, kid. It'll sell a lot of papers." This indicates an expectation that the story will boost sales, but the transcript concludes before any information about the actual sales figures *after* this revelation is provided. Therefore, the statement that sales *increased* after the revelation cannot be verified from the given text. **FALSE** *(Ground-truth: TRUE)*

**Prediction on Fib**: The transcript ends with Mark Chapman, the editor, being apprehended as the killer. His last words are, "Write it up big, kid. It'll sell a lot of papers."This is a prediction about future sales, not a statement of what actually happened to the sales. The transcript provides no information about the newspaper's sales figures *after* it was revealed that the editor was the killer. Therefore, the statement cannot be confirmed as true based on the provided text. **FALSE** *(Ground-truth: FALSE)*

---

Table 13: **Illustrative Example 3 from "Scandal Sheet"**. The pair of claims relies primarily on visual information. Gemini predicts them incorrectly when given only a single modality, but when provided with both the transcript and the video, it correctly identifies the pair of claims.

---

**Example 3 (Part 2) - Movie:** *"Scandal Sheet"*

---

*Video only setting*

**Prediction on Fact**: The video shows a chart of the "NET PAID CIRCULATION NEW YORK EXPRESS" at various points. Initially, at 0:10, the circulation is shown with an upward trend. After the editor is revealed to be the killer and is subsequently shot (around 2:26-2:28), the final shot of the movie (2:29) shows a newspaper headline "KILLER!!" with the byline "By Steve McCleary" superimposed over the same circulation chart. At this point, the line on the chart indicating sales has reached its highest point, implying a significant increase in sales coinciding with or immediately following the revelation and capture of the killer editor. **TRUE** *(Ground-truth: TRUE)*

**Prediction on Fib**: The movie shows a chart of the "New York Express" net paid circulation at 0:10, indicating an upward trend in sales from October to September, reaching 650,000. Later, after the editor (the man standing by the chart at 0:10) is revealed as the killer and is shot (around 2:27), the final shot at 2:29 shows a newspaper headline "KILLER!!" superimposed over the *same* circulation chart. The numbers and the upward trend on this chart are identical to the one shown at 0:10. The movie does not provide any information or visual evidence suggesting that the newspaper's sales decreased after the editor was revealed as the killer. The chart displayed at the end is the same one that showed increasing sales. **TRUE** *(Ground-truth: FALSE)*

---

*Video w/ subs setting*

**Prediction on Fact**: The statement's truthfulness hinges on whether newspaper sales increased after the revelation that the editor, Mark Chapman, was the killer. 1. At 3:40, Mark Chapman discusses the newspaper's circulation, showing a graph and mentioning that when circulation reaches 750,000, a big bonus will be paid. The graph shows circulation rising, with the last point around June/July showing approximately 620,000-650,000. 2. The revelation that Mark Chapman is the killer unfolds towards the end of the film. 3. At 59:39, the newspaper headline "EXPRESS EDITOR LONELY HEART KILLER!!" by Steve McCleary is shown. 4. Immediately following this (59:45 - 59:49), a large circulation meter for the "N.Y. EXPRESS" is shown. A hand physically moves the needle on the meter, stopping at 750,000. "The End" is then superimposed over this image. This sequence strongly implies that the story revealing the editor as the killer led to the circulation reaching 750,000. Earlier, Chapman had also told McCleary to write the story big because "It'll sell a lot of papers" (59:06). The visual of the circulation hitting 750,000 immediately after the headline about the editor being the killer is the final image before "The End," suggesting this was the peak sales driven by this sensational story. **TRUE** *(Ground-truth: TRUE)*

**Prediction on Fib**: At the end of the movie, after the editor, Mark Chapman, is revealed as the killer and the headline "EXPRESS EDITOR LONELY HEART KILLER!!" is shown (59:40-59:43), the circulation meter for the New York Express is shown. A hand moves the needle to 750,000 (59:45). Earlier in the film (around 4:24), it was mentioned that reaching 750,000 in circulation would trigger a big bonus. The film ends with the circulation at this high number, indicating that sales reached this target *after* the revelation about the editor. There is no indication that sales decreased; in fact, the implication is that this sensational story would boost sales, as McCleary says, "Write it up big, kid. It'll sell a lot of papers." (59:14-59:16). **FALSE** *(Ground-truth: FALSE)*

---

Table 14: **Illustrative Example 4 from "Tight Spot"**. Gemini fails to predict both claims correctly when given only the subtitles or only the visual cues. However, when provided with both modalities together, it correctly predicts both claims.

---

**Example 4 (Part 1) - Movie: *"Tight Spot"***

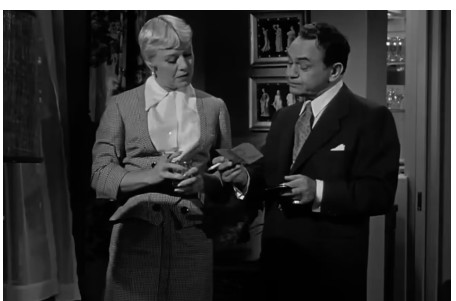 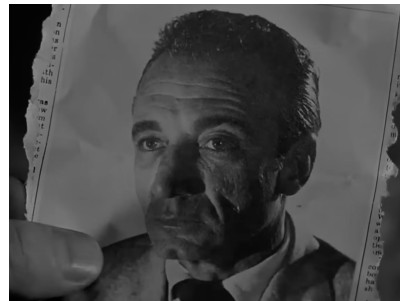

**Fact (✓)**: When Hallet shows Conley a photograph of an individual who was on the yacht, she immediately recognizes the person.
**Fib (✗)**: When Hallet shows Conley a photograph of an individual who was on the yacht, she claims not to recognize the person.

**Granularity**: Single-scene

**Context (Spoiler alert !):** This pair of claims to be answered correctly requires *both* the *visual* and *text* cues. Hallet shows Conley a photograph (visual information) and asks whether she recognizes the person (Senior Maracelli). Conley recognizes the person. *When the model is given only the subtitles, it misclassifies both the fact and the fib (as visual cues are absent). In contrast, when provided with both the video and the subtitles, it predicts each one correctly.*

**Gemini's Predictions:**
*Subs only setting*

**Prediction on Fact**: The statement claims that "When Hallet shows Conley a photograph of an individual who was on the yacht, she immediately recognizes the person." We need to check two things in the transcript: 1. Does Hallett show Conley a photograph of an individual from the yacht? 2. If so, does Conley immediately recognize the person? In the transcript, the relevant exchange is: HALLETT: "And at one of these islands, this man came aboard the yacht." CONLEY: "Signor Maracelli. The Italian garden expert. He taught me to say his name himself." [...] *The transcript does not use the word "photograph" or describe the action of showing one in this specific interaction.* Since the statement's premise that Hallett shows Conley a photograph is not explicitly confirmed by the transcript, a part of the statement is not verifiably true based on the provided text. **FALSE** *(Ground-truth: TRUE)*

**Prediction on Fib**: When Mr. Hallett mentions Signor Maracelli coming aboard the yacht, Conley immediately identifies him as "The Italian garden expert" and adds, "He taught me to say his name himself." This indicates she clearly recognizes the person Hallett is referring to (and presumably showing a photograph of, though the act of showing the photo isn't explicitly stated, Hallett's line "this man came aboard the yacht" implies he's identifying someone specific). She does not claim not to recognize him. **FALSE** *(Ground-truth: FALSE)*

---

Table 15: **Illustrative Example 4 from "Tight Spot"**. Gemini fails to predict both claims correctly when given only the subtitles or only the visual cues. However, when provided with both modalities together, it correctly predicts both claims.

---

**Example 4 (Part 2) - Movie: *"Tight Spot"***

---

*Video only setting*

**Prediction on Fact**: The statement asks whether Conley immediately recognizes the person in the photograph shown to her by Hallet. In the video, at 00:30, Hallet takes out a photograph. At 00:31, he shows it to Conley. Conley looks down at the photo, then at 00:32, she looks up at Hallet with an expression that seems more contemplative or questioning rather than one of instant, clear recognition. She then glances back at the photo before looking at Hallet again and starting to speak at 00:33. Her reaction involves a pause and a sequence of looks, which does not convey *immediate* recognition. "Immediate" suggests an instantaneous, reflexive understanding, which is not evident in her more measured response. **FALSE** *(Ground-truth: TRUE)*

**Prediction on Fib**: The statement claims that when Hallet shows Conley a photograph of an individual from a yacht, Conley claims not to recognize the person. Observing the video, there is a scene from approximately 0:29 to 0:36 where a man (presumably Hallet, played by Edward G. Robinson) shows a photograph to a woman (presumably Conley, played by Ginger Rogers). The woman looks at the photograph (0:31-0:32). However, the video segment does not include any audible dialogue or clear visual cue (like shaking her head definitively or mouthing "I don't know") from the woman that would confirm she "claims not to recognize the person." Her facial expression is somewhat pensive or surprised but not an explicit denial of recognition. She then proceeds to make a phone call (0:34). The context that the photograph is of "an individual who was on the yacht" is also not explicitly provided by the visual information in this clip. Since the crucial part of the statement - her claiming not to recognize the person - is not depicted or audible in the provided video evidence, the statement cannot be verified as true based on the available information. **FALSE** *(Ground-truth: FALSE)*

*Video w/ subs setting*

**Prediction on Fact**: When Hallett shows Conley the photograph (around 00:40), he identifies the man as "Signor Maracelli." Conley immediately confirms recognition by saying, "The Italian garden expert. He taught me to say his name himself." This demonstrates she knew who the person was. **TRUE** *(Ground-truth: TRUE)*

**Prediction on Fib**: When Mr. Hallett mentions Signor Maracelli coming aboard the yacht, Conley immediately identifies him as "The Italian garden expert" and adds, "He taught me to say his name himself." This indicates she clearly recognizes the person Hallett is referring to (and presumably showing a photograph of, though the act of showing the photo isn't explicitly stated, Hallett's line "this man came aboard the yacht" implies he's identifying someone specific). She does not claim not to recognize him. **FALSE** *(Ground-truth: FALSE)*

---

Table 16: **Illustrative Example 5 from "Tight Spot"**. Gemini fails to predict the pair of claims correctly, in every modality setting.

---

**Example 5 (Part 1) - Movie:** *"Tight Spot"*

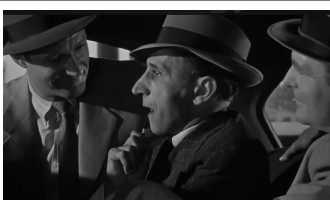 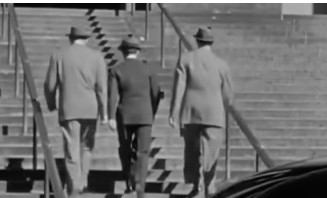 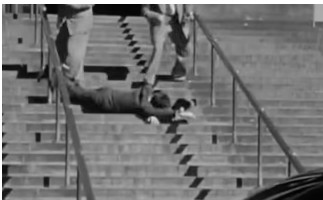

Pete in taxi       Walking the stairs of the church       Pete being shot

**Fact** (✓): Pete Tinelli was shot.
**Fib** (✗): Pete Tinelli was poisoned.

**Granularity**: Multi-scene

**Context (**Spoiler alert !**)**: Early in the film, we are introduced to Pete during a scene in which he rides in a taxi with two other men. Later, the group arrives in front of a church, and as they walk up the steps, Pete is suddenly shot while the other two men look around, trying to identify the shooter. *This pair of claims requires both visual and text cues to be resolved correctly, as the model needs to understand who is Pete (in the first scene) and connect it with his murder (being shot). Gemini fails to predict correctly the pair of claims in every modality setting.*

**Gemini's Predictions:**
*Subs only setting*

**Prediction on Fact**: The transcript states that "Pete's dead" and that "He was murdered by the Costain mob." [..] However, the transcript does not specify the method of murder. It does not explicitly say that Pete Tonelli was shot. Therefore, because the specific detail of "shot" is not confirmed in the transcript, the statement is false. **FALSE** *(Ground-truth: TRUE)*

**Prediction on Fib**: The transcript states, "Pete's dead... He was murdered by the Costain mob." It does not specify the method of murder. Since the transcript doesn't mention poisoning as the cause of Pete Tonelli's death, and only confirms he was murdered, the statement that he was poisoned cannot be confirmed as true based on the provided text. **FALSE** *(Ground-truth: FALSE)*

*Video only setting*

**Prediction on Fact**: A man is visibly shot around the 1:57-1:58 mark in the video. In this scene, a man is held by two others, another man (resembling Edward G. Robinson) nods, a gunshot sound is heard, and the man being held slumps. However, throughout the provided video clip, there is no dialogue, on-screen text, or any other information that identifies the man who was shot as "Pete Tinelli." [...] **FALSE** *(Ground-truth: TRUE)*

**Prediction on Fib**: The video contains several scenes that lead to the conclusion that Pete Tinelli was poisoned. At 1:58, a group of men are shown in what appears to be a garage, and one man is forcibly made to drink something. He subsequently collapses at 2:01. Later, at 2:05, a woman asks, "What happened to Pete Tinelli?" and is told, "He won't bother you anymore." Shortly after, at 2:08, one man informs another, "Pete Tinelli's been taken care of." When asked "How?", the response is "Poison." This sequence of events and dialogue directly confirms that Pete Tinelli was poisoned.**TRUE** *(Ground-truth: FALSE)*

---

Table 17: **Illustrative Example 5 from "Tight Spot"**. Gemini fails to predict the pair of claims correctly, in every modality setting.

---

**Example 5 (Part 2) - Movie:** *"Tight Spot"*

---

*Video w/ subs setting*

**Prediction on Fact**: The provided transcript states at 00:54-00:55 that "Pete's dead" and "He was murdered by the Costain mob." However, neither the transcript nor the accompanying video footage specifies the method of his murder. The statement "Pete Tinelli was shot" asserts a specific cause of death (being shot) which is not explicitly mentioned or shown in the provided materials. Since the method of murder is not confirmed, we cannot ascertain that he was shot based solely on the given information. **FALSE** *(Ground-truth: TRUE)*

**Prediction on Fib**: The transcript states at 00:42, "He was murdered by the Costain mob." While this confirms Pete Tonelli was murdered, it does not specify the method of murder. The statement claims he was poisoned. Since the available information does not mention poisoning as the cause of death, the specific claim that he was poisoned cannot be verified as true from the provided movie and transcript. **FALSE** *(Ground-truth: FALSE)*

---

