# OpenReview forum: "Movie Facts and Fibs (MF$^2$): A Benchmark for Long Movie Understanding"
_ICLR.cc/2026/Conference — Submitted to ICLR 2026_

### Official Review · Reviewer_KDA2 · 2025-10-30

**Soundness:** 3
**Presentation:** 4
**Contribution:** 3
**Rating:** 6
**Confidence:** 5

**Summary:**

This paper introduces MF^2 (Movie Facts and Fibs), a new benchmark designed to evaluate the holistic narrative comprehension of Vision-Language Models (VLMs) across full-length movies, which typically last between 50 and 170 minutes. The authors argue that existing benchmarks fail to assess genuine understanding, instead focusing on narrow retrieval tasks that resemble a "needle-in-a-haystack" approach. MF2 features over 850 manually constructed contrastive claim pairs (a true "fact" and a plausible "fib") for 53 open-licensed films, which require reasoning about core elements like character motivations, causal relationships, and emotional arcs. The evaluation protocol demands that models correctly identify both the true and false claims in a pair, and initial experiments reveal a significant performance gap between state-of-the-art models (Gemini 2.5 Pro) and human evaluators.

**Strengths:**

- The core strength of MF^2 is its dedicated focus on evaluating genuine narrative comprehension rather than shallow retrieval or memorization
- MF^2 contains over 850 manually constructed contrastive claim pairs (fact/fib). This manual effort combats the issues seen in other benchmarks that rely on large-scale, semi-automatically generated questions, which may suffer from model biases
- The dataset uses 53 full-length movies, with an average duration of 88.33 minutes. This scale is longer than most existing video understanding datasets, which often rely on short clips.
- The code and dataset for reproduction are included, which is nice
- The paper is well-written, and the flow is easy to follow

**Weaknesses:**

- Models and humans evaluate each claim independently without knowledge of the paired (fact/fib) structure during prediction. The challenging pairwise accuracy metric (requiring both fact and fib to be correct) is computed after the predictions are made. I don’t see why such a post-hoc setup is necessary
- The strategy used to prevent data contamination, while effective, introduces a bias toward a specific style of filmmaking. Added to the fact that the video quality will not be as good since the movies are very old. I am wondering how much of this is to blame for the low performance of some models.
- The dataset’s evaluation protocol means the task is essentially two separate binary classification tasks, rather than a single task requiring the model to resolve the subtle differences between two highly similar statements. The benchmark thus measures independent truth prediction ability, but not necessarily the model's ability to perform contrastive reasoning based on the textual juxtaposition of the claims themselves

**Questions:**

See weaknesses plus
- Would focusing on older films released between 1920 and 1970 be an issue, since video quality was not as good back then, thus increasing the difficulty of the dataset?
- How is this dataset different from Tropes datasets which also use binary classification and long movies, and which often focus on narrative reasoning?

---

> ### Author Response · Authors · 2025-11-20
> **Answer to reviewer KDA2 (part 1)**
>
> We thank the reviewer for their time and effort spent to evaluate our work, and for recognizing the substantial manual annotation, which we consider key to ensuring the benchmark’s quality and reliability.
>
> >[..] The challenging pairwise accuracy metric (requiring both fact and fib to be correct) is computed after the predictions are made. I don’t see why such a post-hoc setup is necessary. + The benchmark measures independent truth prediction ability, but not necessarily the model's ability to perform contrastive reasoning based on the textual juxtaposition of the claims themselves + the task is essentially two separate binary classification tasks, rather than a single task requiring the model to resolve the subtle differences between two highly similar statements.
>
> Our evaluation protocol is motivated as follows:
>
> Multiple-choice settings introduce distractor-driven biases, where models exploit superficial cues or distractor order [1]. Following prior work on narrative understanding [2], we evaluate each minimally differing claim **independently**, removing distractors and reducing such shortcut behaviors. Under this setup, pairwise accuracy is stricter and more meaningful metric; a model is counted as correct only if it predicts **both** labels accurately, **minimizing the chance that it appears “correct for the wrong reason”**[2].
>
> The task **is equivalent to two independent binary classification tasks only at inference time**. Although we obtain a true/false prediction for each claim independently, the evaluation protocol differs: it is **pairwise**, not item-wise. This makes the protocol different from the standard binary classification evaluation accuracy in which each prediction contributes independently to the score. In our evaluation, **the relationship between the two predictions is essential**. The model must **implicitly distinguish** what truly happened from a minimally edited alternative that is also plausible, even though it never sees the two statements jointly during inference. That said, the evaluation depends on the **relationship between the two predictions**, not the predictions alone.
>
>
> As a further illustration of why interpreting results based only on simple accuracy is insufficient, below we provide the results of Gemini2.5 in 3 settings (video only, subs only, video w/ subs) additionally reporting “Accuracy on facts” and “Accuracy on fibs” for Gemini 2.5. The numbers are shown as:
>
> Modality_used: [Pairwise Accuracy, Accuracy (average), Accuracy on Facts, Accuracy on Fibs]
>
> video_only=[37.21,64.22, 47.58, 80.87]
>
> subs_only=[56.68, 76.15, 61.05, 91.24]
>
> video_and_subs=[60.59, 77.64, 69.12, 86.17]
>
> **Simple accuracy can appear high simply because the model performs well on the fibs, while failing on the facts. This pattern indicates a bias toward predicting False**.
>
> Importantly, **this design does not limit the benchmark’s usability**. Because we release all data and claim pairs, researchers who wish to evaluate contrastive reasoning can reformulate the task into a multiple-choice setup or any other comparative format. Thus, our chosen protocol reflects our evaluation priorities, while the benchmark remains flexible and fully compatible with alternative formulations.
>
> [1] Molfese et al., Right answer, wrong score, arXiv:2503.14996, 2025.
>
> [2] Karpinska et al., One thousand and one pairs, EMNLP 2024.

---

> ### Author Response · Authors · 2025-11-20
> **Answer to reviewer KDA2 (part 2)**
>
> > [..] introduces a bias toward a specific style of filmmaking. Added to the fact that the video quality will not be as good since the movies are very old. I am wondering how much of this is to blame for the low performance of some models.
>
> To be honest, models have likely encountered information about these films during pretraining (e.g., synopses, summaries, discussions), and the Hollywood-style narrative structure is highly familiar[1,2,3]. In principle, such exposure should help rather than hinder performance. Yet despite this potential familiarity, models still lag far behind humans, suggesting **the benchmark demands capabilities that go well beyond general world knowledge or stylistic priors**, underscoring its value (see examples in updated version-Appendix F, L1568-2089).
>
> We also view the trade-off between stylistic invariance and legal openness as acceptable: **fully accessible content avoids reproducibility issues**, whereas many existing benchmarks (e.g., Video-MME, LVBench) rely on YouTube links or keyframes that **often become unavailable over time**.
>
> Regarding video quality, our benchmark does not rely on fine-grained perceptual details (e.g., colors of objects, small props). **The claims target higher-level comprehension that remains perfectly accessible even in older films, as shown by near-perfect human performance.** Thus, **video resolution is unlikely to be the limiting factor in model performance**.
>
>
> In the updated version of the paper, we provide illustrative examples of claims with Gemini’s predictions, showing that the model does reason over visual content, indicating that video quality is not the primary bottleneck (Appendix F, L1568–2089). **Example 3 is particularly relevant**: it requires strong visual understanding to be answered correctly. The attached frames demonstrate that, if video quality was the limiting factor, the model would not succeed on this case.
>
> [1]Cutting, J.E. 2016 Narrative theory and the dynamics of popular movies. Psychon. Bull. Rev.
>
> [2] Patrice Pavis. 1998. Dictionary of the theatre: Terms, concepts, and analysis. University of Toronto Press.
>
> [3]Gustav Freytag. 1896. Freytag’s technique of the drama: an exposition of dramatic composition and art. Scholarly Press.
>
> >How is this dataset different from Tropes datasets which also use binary classification and long movies, and which often focus on narrative reasoning?
>
> We would kindly ask the reviewer to provide specific references to the “Tropes datasets” being referred to, so that we can conduct a direct and comprehensive comparison. Based on our literature search, the works we identified are PicTropes [1], Truman [2], and trope resources such as TVTropes.
>
> In general there are substantial differences between our work and tropes. Particularly:
>
> 1) Tropes focus on thematic, high-level patterns rather than core narrative comprehension. They apply globally across films, and models are asked whether a trope fits an entire movie. In contrast, MF2 contains movie-specific claims targeting central, memorable narrative events. Answering each pair correctly requires understanding **character motivations, emotions, causal links, and event order, not broad thematic patterns**.
>
> 2) Tropes datasets are built largely from text metadata (e.g., DBTropes.org, TVTropes summaries), and models can often succeed by recalling such metadata or pretrained textual associations. In contrast, our benchmark is explicitly designed to require **both visual and linguistic cues to answer the claims correctly** (see examples in the updated version; Appendix F).
>
> 3) Each claim pair (fact vs. fib) is minimally edited and both versions remain plausible. This forces models to identify what actually occurred, rather than judging global fit or thematic likelihood as done in Tropes.
>
> 4) Our videos are full-length movies, unlike prior trope datasets with extremely short animation clips like Truman[2].
>
> **These distinctions highlight that, although both settings involve films the underlying tasks, reasoning skills, and dataset construction philosophies are fundamentally different**.
>
> [1]García-Ortega, R. (2018). Overview of PicTropes, a film trope dataset. arXiv.
>
> [2]Su, Hung-Ting, et al. "Truman: Trope understanding in movies and animations." Proceedings of the 30th ACM International Conference on Information & Knowledge Management. 2021.
>
> **Closing Note:**
>
> We thank the reviewer once again for the time spent evaluating our work. We hope that the extended clarifications regarding our design choices address your concerns and will allow you to revisit your score. We remain fully open to any further questions, suggestions, or discussion.

---

> ### Comment · Reviewer_KDA2 · 2025-11-22
>
> I want to thank the authors for their thorough answers to my concerns and questions. The presented arguments make sense to me, and I appreciate the added experiments, such as the "accuracy on Facts/Fibs" (I would recommend the authors to also include these in the Appendix) and Appendix F.
>
> I am thus raising my score from 6 to 8.

---

> > ### Author Response · Authors · 2025-11-24
> >
> > We sincerely thank the reviewer for their follow-up and for the positive reassessment of our work. We will include the results in the next updated version (we are currently awaiting feedback from the remaining reviewers).

---

### Official Review · Reviewer_rdkg · 2025-10-30

**Soundness:** 3
**Presentation:** 3
**Contribution:** 2
**Rating:** 4
**Confidence:** 3

**Summary:**

The paper presents a Q&A benchmark for long movie understanding. The key novelty of the benchmark is that each Q (850 in total) has two possible answers, one true (fact) and one false (fib) and the goal is to correctly understand both the true and false answers.

**Strengths:**

- The paper touches a very important part in movie understanding, i.e. the ability of MLLMs to understand true facts from counterfactual events that could be plausible or sound realistic. I think this is a very nice ability for the MLLMs to account for.

- The paper is well-written and easy to follow.

- The work offers high-quality human annotations (not automatically done).

**Weaknesses:**

There are several drawbacks associated with the work.

1/ Evaluation

1i/ Subtitles only.
The paper analyses how much subtitles help visual information; however, most recent studies have shown the big modality gap in these MLLMs, which begs the question whether the proposed questions can be answered solely with subtitles. Table 4 further supports this: it shows that subtitles are the ones providing the most information to reach the target performance. In this case, why is video and generally visual information needed?

1ii/ Pairwise accuracy metric.
Although I appreciate proposing a new metric and understand that it’s hard to convince the audience, the pairwise accuracy seems weird and is somehow tailored to artificially lower performance. It is unclear what this metric offers more than simply accuracy.

2/ Experimental setup with the pairwise accuracy computation.
Given this metric, it is unclear how logits are being transformed into pairwise accuracy. To my understanding, each option in the Fib & Fact is treated independently. However, I think it would be essential to have more evaluations with different assignments. Some examples, given that only one statement is true in every answer (and one is false), are:
(i) if the first one is assigned true, the second should be assigned as false (and vice versa);
(ii) assign the true or false based on the logits;
(iii) ask the MLLM if it is certain of its choice.
This (non-exhaustive) list may help us see whether existing VLMs actually confuse fibs from facts in this setting.

3/ The examples in the sup mat show that aquarium vs bus or appeared happy vs appeared bothered.
These counterfactuals are visible by single frames or even simply by text. It is unclear why both modalities are needed. A good way to evaluate this would be to remove the word completely (instead of changing it) and ask the VLM to predict it (this or a synonym), similar to “fill the gap” part.

4/ Inconsistent results between Table 3 and Table 4.

5/ Data contamination.
Table 4 shows the zero shot performance to be 66.5% with only movie titles! This clearly shows that there is a strong data leakage. This suggests that no matter the design of evaluation protocols and confusion, the models have seen the data. This makes it hard to know whether performances come from models, params or simply data leakage.

6/ Limited evaluation protocol.
Evaluating on multiple-choice QA is limited compared to open-ended or generative movie understanding tasks. It is unclear whether performance correlates with genuine scene-level understanding.

7/ Data diversity.
Given that all movies come from the 20th century, this limits the diversity of the data.

8/ Dataset scale.
The dataset contains 53 movies and approx. 850 Q&A pairs, which makes it small compared to modern multimodal datasets. The scale together with the limited diversity may compromise any drawn conclusions as a model may overfit to these data/domains without actual generalization capabilities.

Minor comment.
Given English subtitles, it would be interesting to have a discussion about language bias.

**Questions:**

For questions, it would be great if the authors could address some points from the weaknesses above. Specifically:

Q1 (W1i) It would be useful to have performances by using only subtitles.

Q2 It would be great if the authors showed the need for visual understanding. One way to do this would be by examining whether models actually use the visual stream vs mostly (or solely) the textual one, for example, by using counterfactual visual cues or by shuffling scenes.

Q3 (W1ii, W2) Pairwise accuracy metric.

Q4 It would be great if the authors could provide some failure case analysis, showing reasoning errors or multimodal misalignments, especially to show the need for vision.

Q5 (W5) Given the data contamination observed in Table 4, it would be beneficial if the authors could test this by using fictional or synthetic movie titles or shuffling titles or reusing parts from other movie titles?

Q6 (W7, W8) It would be great if the authors can explain how to prevent overfitting, especially if the dataset is intended for leaderboard-style competition.

Q7 (minor) It would be useful to check if model predictions correlate with the number of frames sampled per movie, for example, by examining if more visual input (ie more frames) improves reasoning?

---

> ### Author Response · Authors · 2025-11-20
> **Answer to reviewer rdkg (part 1)**
>
> We thank the reviewer for their time and effort, and for recognizing the substantial manual annotation, which we consider key to ensuring the benchmark’s quality and reliability.
>
> >Pairwise accuracy metric. [...] artificially lower performance. It is unclear what this metric offers more than simply accuracy. + 2/ [..] Given this metric, it is unclear how logits [..]
>
> There seems to be a misunderstanding regarding our inference procedure and how pairwise accuracy is computed. We clarify below why this metric is more appropriate than simple accuracy.
> We run inference **independently on each claim**. Open-weight models are prompted to generate only “true” or “false,” while closed-source models provide an explanation followed by a final label. **We extract this label using standard regex-based parsing [1,2]; logits are never used. Pairwise accuracy is then computed by marking a (fact, fib) pair as correct only if both statements are classified correctly**.
>
> **Why do we choose pairwise-accuracy over “simple accuracy”?**
> Multiple-choice settings introduce distractor-driven biases, where models exploit superficial cues or distractor order [1]. Following prior work on narrative understanding [2], we evaluate each minimally differing claim **independently**, removing distractors and reducing such shortcut behaviors. Under this setup, pairwise accuracy is stricter and more meaningful metric; a model is counted as correct only if it predicts **both** labels accurately, **minimizing the chance that it appears “correct for the wrong reason”**[2]. In contrast, in simple accuracy each prediction contributes independently to the score making it less suitable for our case and more suitable for multiple-choice formats, which we intentionally avoid for the aforementioned reasons.
>
> As a further illustration of why interpreting results based only on simple accuracy is insufficient, below we provide the full version of Table 4, additionally reporting “Accuracy on facts” and “Accuracy on fibs” for Gemini 2.5. The numbers are shown as:
>
> Modality_used: [Pairwise Accuracy, Accuracy (average), Accuracy on Facts, Accuracy on Fibs]
>
> movie_title=[43.77, 66.30, 50.46, 82.14], synopsis = [25.46, 61.80, 26.95, 96.65]
>
> video_only=[37.21,64.22, 47.58, 80.87], subs_only=[56.68, 76.15, 61.05, 91.24]
>
> video_and_subs=[60.59, 77.64, 69.12, 86.17]
>
> Simple accuracy can appear high simply because the model performs well on the fibs, while failing on the facts. Models are also biased towards predicting False.
>
> [1] Molfese et al., Right answer, wrong score, arXiv:2503.14996, 2025.
>
> [2] Karpinska et al., One thousand and one pairs, EMNLP 2024.
>
> >Subtitles only. [...] begs the question whether the proposed questions can be answered solely with subtitles. [..] why is video and generally visual information needed? + Q1 It would be useful to have performances by using only subtitles.
>
> Narrative structure in film relies on linguistic, auditory, and visual cues—conversations, music, scene transitions—that collectively shape the plot [1,2]. **Because subtitles omit many of these elements, they cannot capture the full storyline**. To reflect this multimodal structure, annotators were instructed to create grounded claims that using **both linguistic and central visual aspects of the narrative (not peripheral details)**. As a result, some claims require both modalities and cannot be answered using subtitles alone. **We provide examples along with Gemini’s explanations and predictions in the updated version of the paper.** (Appendix F, L1568-2089).
>
> Our analysis on Gemini 2.5 (Table 4) shows that **visual information is not redundant: adding video yields an improvement in pairwise accuracy**. The modest gains likely reflect **current VLM limitations, such as strong reliance on language over visual information [3] or difficulty with long contexts [4,5], rather than the irrelevance of visual cues for the task**.
>
> **Following your suggestion**, we also ran the “subtitles only” setting for two open-weight models. We report [Pairwise Accuracy, Average Accuracy, Accuracy on Facts, Accuracy on Fibs] InternVL3-78B achieves [**48.96**, 72.01, 59.91, 84.10] while Qwen2.5-VL-72B [**37.55**, 66.76, 41.12, 92.40].
>
> These results are below those obtained with video+subs (Table 3), indicating again that visual information is not irrelevant.
>
> [1]Cutting, J.E. 2016 Narrative theory and the dynamics of popular movies. Psychon Bull Rev
>
> [2] Patrice Pavis. 1998. Dictionary of the theatre: Terms, concepts, and analysis. University of Toronto Press
>
> [3]S. Fu et al., Hidden in plain sight, COLM 2025
>
> [4]Nelson F. Liu et al., Lost in the Middle, TACL 2024
>
> [5] Hosseini P. et al.,  Efficient Solutions For An Intriguing Failure of LLMs, COLING 2025

---

> ### Author Response · Authors · 2025-11-20
> **Answer to reviewer rdkg (part 2)**
>
> >Some examples, [..] (ii) assign the true or false based on the logits; (iii) ask the MLLM if it is certain of its choice. [..] see whether existing VLMs actually confuse fibs from facts in this setting.
>
> We follow your (iii) suggestion , and we prompt the models for certainty scores. In particular, we prompt the model to provide a response (true or false) and a confidence score (0-100). If the confidence is <50% we flip the answer. We run this experiment on Qwen2_5VL-72b and the results are exactly the same. The model is highly confident on the predictions (with no flips). The experiment was conducted on the video+subs setting. Let us know if you want to include this experiment on a next updated version of the paper.
>
> Nonetheless, the answer extraction method we use via regex, is a widely adopted and model-agnostic practice[1,2], that other similar datasets use as well.
>
> [1] Wang et al., LVBench, arxiv 2024
>
> [2] Fu et al. , Video-MME arxiv 2024
>
> > Q2 It would be great if the authors showed the need for visual understanding. One way to do this [..] using counterfactual visual cues or by shuffling scenes. + Q4 It would be great if the authors could provide some failure case analysis
>
> We followed your suggestion and ran an experiment by shuffling the frames for the same model, Qwen2_5VL-72b (due to budget constraints, we are not able to run this for Gemini) The results further support that visual information are actually used  (**results are also included in the updated version of the paper Appendix E, Table 6**) :
>
> We report the results in the format: Model / Modality = [Pairwise Accuracy, Accuracy, Accuracy on Facts, Accuracy on Fibs]
>
> Vision+subtitles / Qwen2.5-VL-72B= [45.85, 70.39, 55.65, 85.14]
>
> Vision(shuffled) +subtitles / Qwen2.5-VL-72B= [40.32, 66.19, 46.08, 86.29 ]
>
> Vision only  /  Qwen2.5-VL-72B= [29.72, 58.76 , 45.39, 72.12]
>
> Vision only (shuffled)  /  Qwen2.5-VL-72B= [4.84, 51.84 , 4.95, 98.73]
>
>
> Nevertheless, whether current models **actually use visual details** is separate from whether **visual information is needed to resolve the claims**. To further address both of your points, we provide examples in the updated version of the paper (Appendix F) that require visual evidence along with Gemini’s predictions (**showing also failure cases**), which explicitly reference specific scenes (indicating that visual input is indeed being used for Gemini). The real challenge is whether models can use visual and textual streams **effectively over very long contexts**, which explains why their performance remains far below human level.
>
> >3/ The examples in the sup mat show that [..] remove the word completely (instead of changing it) and ask the VLM to predict it (this or a synonym), similar to “fill the gap” part.
>
> We understand that the examples you are referring to might seem solvable by text or a single frame; however, without access to the full movie or additional context, it is difficult to make a definitive judgment. For this reason, **we provide examples on the updated version of the paper (see Appendix F), giving additional context to clarify why it is not solvable by a single frame or by a single modality and we provide the predictions of Gemini as well.** For the specific example you asked (aquarium vs bus), it is not solvable by a single frame as the viewer needs to understand correctly what happened in the aquarium (if a bomb exploded there) and in the bus (see similar examples in the updated version of paper).
>
> While the “fill-the-gap” idea is interesting, implementing it is less straightforward compared to the evaluation setting we use. Claims while following minimal-edits, might differ in more that one word and multiple valid completions could exist, making this evaluation setting ambiguous/less straightforward to use.
>
> > 4 Inconsistent results between Table 3 and Table 4.
>
> Indeed there was a typo in Table3: the accuracy of Gemini2.5 for the video+subs setting is 77.64%, which we round to 77.6%.  Thank you for pointing it out, we updated it.
>
> > 5/ Data contamination. Table 4 shows the zero shot performance to be 66.5% with only movie titles! [..] This makes it hard to know whether performances come from models, params or simply data leakage.
>
> We agree that some degree of data leakage is likely (e.g., exposure to related synopses, reviews, or metadata during pretraining; Section 4.2). However, controlling for this is practically impossible given the massive and diverse nature of pretraining data. **Importantly, even if Gemini-2.5 has seen parts of this data, its performance remains well below human-level accuracy, indicating that the task continues to be challenging**.

---

> ### Author Response · Authors · 2025-11-20
> **Answer to reviewer rdkg (part 3)**
>
> >6/ Limited evaluation protocol. Evaluating on multiple-choice QA is limited [..]
>
> We assess models independently for each claim using a binary evaluation protocol, not as multiple-choice QA. Multiple-choice formats suffer from distractor and order biases, while LLM-based judging is unreliable for long narratives, as shown by frontier models failing to understand extended videos[1,2,3].
>
> [1]Ye et al. , Justice or Prejudice? Quantifying Biases in LLM-as-a-Judge. , ICLR 2025
>
> [2]Liu & Zhang, Is Your Video Language Model a Reliable Judge?, ICLR 2025
>
> [3]Bavaresco et al. , LLMs instead of Human Judges? … , ACL 2025
>
> >“It is unclear whether performance correlates with genuine scene-level understanding”
>
> We do not target scene-level visual details but comprehension of the core storyline. For humans, we believe it is inaccurate to suggest that performance is unrelated to genuine understanding: **humans cannot rely on shortcuts or metadata and must interpret both language and visuals to answer the claims correctly, which are designed to require such understanding**. Near-perfect human accuracy therefore supports the task’s validity. For models, this concern is less central: their much lower accuracy already shows they lack this type of long-context narrative understanding.
>
> >7/ Data diversity. Given that all movies come from the 20th century, this limits the diversity of the data.
>
> The decision to use public-domain films was a **trade-off to ensure legal openness, reproducibility, and redistribution**.  While these films reflect the stylistic norms of their historical period, they span a wide **range of genres across five decades**, which does not necessarily imply cultural poverty or homogeneity despite their earlier era. We consider this trade-off **acceptable because fully accessible content avoids the reproducibility issues of datasets** relying on YouTube links or keyframes (e.g., Video-MME, LVBench), which often **become unavailable over time**. Although these films follow predominantly Hollywood-standard narrative structures, current systems still fall far below human performance, highlighting the **benchmark’s difficulty despite reduced stylistic variation**.
>
> Importantly, our goal is **not** to claim that solving this benchmark would amount to solving video reasoning as a whole. Rather, we argue that the skills evaluated here are **necessary but not sufficient** for a model to qualify as a good video reasoning system. We are not suggesting that a model that performs well on MF2 has mastered video reasoning; instead, the substantial gap between human and model performance reveals a clear weakness in current systems (and also a gap in current evaluation frameworks). A human with no prior exposure to old films can still answer these questions with high accuracy, whereas current models struggle significantly. **These core narrative-understanding abilities remain essential regardless of a film’s era or stylistic traits.** Diversity becomes a concern only if the benchmark is interpreted as a **sufficient test of video reasoning, which we do not claim.**
>
> >8/ Dataset scale. [..] The scale together with the limited diversity may compromise any drawn conclusions as a model may overfit [..]
>
> **We think there is a misunderstanding about the nature of this dataset. It is only intended for evaluation, not training, therefore there is no risk of overfitting**. Regarding the size of it, we choose fully manual annotation to ensure high quality data, prioritizing accuracy over scale, a common trade-off in benchmarks (even though benchmarks that are fully manually annotated also follow similar sizes [1,2,3]).
>
> [1] Dong et al., "Benchmarking and Improving Detail Image Caption", arxiv 2024
>
> [2] Hu et al. , Video-MMMU arXiv e-prints 2025
>
> [3]Zhou et al. , "Instruction-Following Evaluation for Large Language Models," arXiv 2023
>
>
> >It would be great if the authors can explain how to prevent overfitting, especially if the dataset is intended for leaderboard-style competition.
>
> The benchmark’s long-term utility is protected because we can always release a Version 2 with entirely new fact/fib pairs, potentially from the same videos but with held-out labels for leaderboard settings. Even if models train on the current release, they would likely not generalize to new fact/fib pairs. While future leaderboard challenges may restrict label access, in this work we release labels to provide a high-quality, open, copyright-free benchmark that encourages research on long-form movie understanding and memory consolidation.
>
> **Closing Note:**
> We thank the reviewer for the time spent evaluating our work. We hope that the extended clarifications regarding our design choices address your concerns. We have also extensively followed your suggestions regarding additional experiments.  We sincerely hope these points will allow you to revisit your score. We remain fully open to any further questions, suggestions, or discussion.

---

> > ### Author Response · Authors · 2025-11-25
> >
> > Dear reviewer, as the discussion period ends in about a week (December 3rd), we are following up to check whether our clarifications and additional experiments addressed your concerns. We would appreciate your engagement in further discussion or a reconsideration of your assessment if the new information resolves the issues you raised.

---

### Official Review · Reviewer_Nok5 · 2025-11-01

**Soundness:** 2
**Presentation:** 2
**Contribution:** 2
**Rating:** 4
**Confidence:** 3

**Summary:**

This paper introduces MF2, a novel benchmark designed to evaluate holistic narrative understanding in full-length movies (50-170 minutes) by requiring models to distinguish between manually crafted fact-fib claim pairs targeting core story elements like character motivations and causal chains. The benchmark includes 53 open-licensed films and over 850 contrastive pairs, adopting a binary evaluation protocol to reduce biases associated with multiple-choice formats and emphasize reasoning over retrieval.

**Strengths:**

The benchmark's use of open-licensed content and human-annotated claims ensures reproducibility and high-quality labels, addressing gaps in existing datasets prone to copyright issues or automated generation. The contrastive claim design and granular categorization (e.g., single-scene to global reasoning) enable a nuanced assessment of narrative comprehension, with human baselines highlighting the task's feasibility for humans but difficulty for models.

**Weaknesses:**

1.	The movies in MF2 are exclusively from 1920–1970 (to avoid data contamination), lacking modern films with contemporary narrative styles, visual effects, or cultural contexts. This limits the generalization of results to real-world scenarios involving recent long videos (e.g., modern films, documentaries). It is suggested that the authors discuss this limitation.
2.	All annotators are co-authors of the paper, rather than independent external annotators. This may introduce subjective biases in claim design (e.g., consistent preferences for certain narrative elements) and quality control, weakening the objectivity of the benchmark.
3.	The experiments only evaluate a narrow set of mainstream VLMs (e.g., Gemini 2.5 Pro, InternVL3). This fails to test the benchmark’s adaptability to diverse model architectures. It is recommended that the authors follow LVBench and evaluate more VLMs.

**Questions:**

NA

---

> ### Author Response · Authors · 2025-11-20
> **Answer to reviewer Nok5 (part 1)**
>
> We thank the reviewer for the time and effort spent to evaluate our work. We also greatly appreciate your recognition of the high-quality labels and contrastive claim design, which we believe are central to the benchmark’s contribution.
>
> >The movies in MF2 are exclusively from 1920–1970 (to avoid data contamination), lacking modern films with contemporary narrative styles, visual effects, or cultural contexts. This limits the generalization of results [...]. It is suggested that the authors discuss this limitation.
>
> We thank the reviewer for this observation. The use of public-domain films (1920–1970) was a **deliberate choice** to ensure **legal openness, reproducibility, and long-term accessibility**, as contemporary films are usually under copyright. While these films reflect the stylistic norms of their historical period, potentially limiting generalization to culturally diverse media, they span a wide **range of genres** across five decades, which does not necessarily imply cultural poverty or homogeneity despite their earlier era. We consider this trade-off acceptable given the benefit of **fully accessible and copyright-free content**, unlike datasets  (e.g., Video-MME, LVBench) that rely on YouTube links or keyframes, which become unavailable over time, **risking long-term reproducibility**. Moreover, although the selected films follow predominantly Hollywood-standard storytelling conventions (often characterized by familiar act structures that one might expect models to handle well [1,2,3]) current systems still fall substantially short of human performance. This gap underscores the **difficulty of our benchmark**.
>
> Finally, our goal is **not** to claim that solving this benchmark would amount to solving video reasoning as a whole. Rather, we argue that the skills evaluated here are **necessary but not sufficient** for a model to qualify as a good video reasoning system. We are not suggesting that a model that performs well on MF2 has mastered video reasoning; instead, the substantial gap between human and model performance reveals a clear weakness in current systems (and in evaluation frameworks). A human with no exposure to old films can still answer these questions with high accuracy, whereas current models struggle. **These core narrative-understanding abilities remain essential regardless of a film’s era or stylistic traits.** Stylistic diversity becomes a concern only if the benchmark is interpreted as a **sufficient test of video reasoning, which we do not claim.**
>
> [1]Cutting,J.E.Narrative theory and the dynamics of popular movies. Psychonomic Bulletin & Review,2016
>
> [2]Pavis, P. Dictionary of the Theatre:Terms, Concepts, and Analysis.University of Toronto Press,1998
>
> [3]Freytag, G. Freytag’s Technique of the Drama.1896
>
> >All annotators are co-authors of the paper, rather than independent external annotators [...]
>
> We did not involve external crowdsourced annotators or semi-automatic approaches, as constructing claims targeting central narrative events, memorable for humans and often requiring reasoning across multiple scenes, is highly challenging. Semi-automatic methods with limited oversight would introduce errors and conflict with our motivation to focus on events that are genuinely memorable for humans. Using crowdsourced annotators would have required **substantial additional time for training and guidance, increased costs, and risked lower annotation quality**.  On the contrary, the setup of authorship particularly **motivates highly qualified contributors** (e.g., PhD students) who would not typically undertake such long-form video annotation as paid crowdwork. For these reasons, we deliberately invested the effort to fully manually construct the dataset.
>
> Regarding diversity, **the annotation process involved 26 annotators, all of whom are co-authors of this work, from 12 institutions in 7 countries**, including PhD students, postdocs, and faculty members. We believe that this mix of backgrounds  contributes to balanced, high-quality claims rather than weakening objectivity.  In addition to providing detailed annotation guidelines(Appendix L119-1504, updated version), **we conducted multiple rounds of discussions and collective revisions to refine annotations**, further strengthening the benchmark’s consistency and rigor. Although annotators knew that the project aimed to build a benchmark, **they were not informed of any specific hypotheses, expected findings, or model-related goals** during annotation, ensuring that annotations were not biased toward any particular outcome.
>
> **We respectfully argue that all the aforementioned factors meaningfully reduce the risk of subjective biases. This practice is also aligned with several established benchmarks [1,2], that similarly rely on co-author annotators**. This level of collaborative involvement reinforces the benchmark’s overall consistency.
>
> [1] Romanou et al., INCLUDE, ICLR 2025.
>
> [2] Enevoldsen et al., MMTEB, ICLR 2025.

---

> ### Author Response · Authors · 2025-11-20
> **Answer to reviewer Nok5 (part 2)**
>
> >The experiments only evaluate a narrow set of mainstream VLMs (e.g., Gemini 2.5 Pro, InternVL3). This fails to test the benchmark’s adaptability to diverse model architectures. It is recommended that the authors follow LVBench and evaluate more VLMs.
>
> We followed your suggestion and we added more models. Importantly, our final set covers a wide range of families/architectures: closed source models, an extension of the original LLaVA model paired with the SigLiP vision encoder (LLava-Video), a version of it trained on CinePile (Video-LLaVA-7B-hf-CinePile), LLaVA-OneVision-1.5 (based on RICEViT vision encoder), VLMs that support long videos (Ovis2, Qwen2.5-VL, InternVL3) and their updated versions that were trained with more multimodal data and RL improvements (Qwen3VL, InternVL3_5, Gemma3). Lastly we experimented with long-video specialized models like Long-Vila-R1, Kangaroo and VideoLLaMA3.
>
> Our final set includes:
>
> -Closed source models:
>
> Gemini2.5-Pro (also in LVBench)
>
> GPT-4o
>
> Claude3.7 (added)
>
> -Open weight models:
>
> VideoLLaMA3(also in LVBench)
>
> Qwen2.5-VL (also in LVBench)
>
> Kangaroo 8B (added, also in LVBench) - (fails to follow the instruction and provide a True/False response, so we exclude it from the results)
>
> InternVL3_5  (added)
>
> Qwen3VL (added)
>
> Gemma3 (added)
>
> LLaVA-OneVision 1.5-7b (added, with an earlier version also in LVBench)
>
> Video-LLaVA-7B-hf-CinePile (added, but fails to follow the prompt and it hallucinates repeating the questions, so we exclude it from the results)
>
> InternVL3
>
> LLaVA-Video
>
> Ovis2
>
> LongVILA-R1
>
> **Results with pairwise and average accuracy are included in the updated version of the paper** (Appendix D, see Table 5). In the final version, where additional space will be allowed, we will move them into the main text. We also attach them below.
>
> Results in format:
>
> Model / Modality = [Pairwise Accuracy, Accuracy, Accuracy on Facts, Accuracy on Fibs ]
>
> Vision+subtitles / InternVL3_5-38B-Instruct = [46.20, 70.33, 61.64, 79.03]
>
> Vision+only / InternVL3_5-38B-Instruct = [26.61 , 59.97, 38.82, 81.11]
>
> Vision+subtitles / Claude3.7 = [44.59, 71.49, 46.31, 96.66]
>
> Vision+only / Claude3.7 = [3.80, 51.44 , 3.92 , 98.96]
>
> Vision+subtitles / Qwen3VL-30B-A3B-Instruct = [42.28, 68.55, 51.50, 85.60 ]
>
> Vision+only / Qwen3VL-30B-A3B-Instruct = [19.24, 56.91, 24.31, 89.52  ]
>
> Vision+subtitles / Gemma3-27B-it = [42.86, 68.09, 60.37, 75.81]
>
> Vision+only / Gemma3-27B-it = [31.46,61.23, 57.37, 65.09]
>
> Vision+subtitles / LLaVA-OneVision 1.5-8b = [26.38, 60.94, 32.03, 89.86]
>
> Vision+only / LLaVA-OneVision 1.5-7b = [4.92, 50.96, 5.88, 96.04]
>
> Kangaroo and Video-LLaVA-7B-hf-CinePile fail to follow the instructions and provide a True/False response.
>
>
> **Closing Note:**
> We thank the reviewer once again for the time spent evaluating our work. We hope that the clarifications regarding our design choices (including the use of public-domain films (1920–1970) to ensure legal openness and reproducibility, the rationale for relying on co-author annotators, and the measures taken to ensure annotation quality) address your concerns. We have also followed your suggestion to substantially extend the experimental coverage. We sincerely hope these points will allow you to revisit your score. We remain fully open to any further questions, suggestions, or discussion.

---

> > ### Author Response · Authors · 2025-11-25
> >
> > Dear reviewer, as the discussion period ends in about a week (December 3rd), we are following up to check whether our clarifications and additional experiments addressed your concerns. We would appreciate your engagement in further discussion or a reconsideration of your assessment if the new information resolves the issues you raised.

---

### Official Review · Reviewer_BnV2 · 2025-11-04

**Soundness:** 2
**Presentation:** 2
**Contribution:** 3
**Rating:** 6
**Confidence:** 3

**Summary:**

This paper introduces MF², a new benchmark designed to evaluate narrative comprehension in long-form movies (average 88 minutes, 53 movies in total).
26 humans watched movies and generated paired statements consisting of an accurate factual description (fact) and a minimally modified false statement (fib).
For each pair, they specified the required scene granularity (single/multi/global) for answering and selected one or more comprehension dimension from {event/entity understanding, temporal perception, emotion understanding, causal reasoning}.
In total, 868 pairs were constructed.
This paper provide the result on evaluated performance using both publicly available recent VLM models and proprietary models such as GPT-4o and Gemini 2.5 Pro, alongside human assessments.
Also, the results of ablation studies are provided to examine the impact of input modality in (video,  subtitle).

**Strengths:**

- Benchmark datasets are valuable assets for our community. The authors commit to releasing the full dataset, code, and movies, ensuring reproducibility and supporting future research.
- Movies are long-form video that mostly self-contained stories. MF² focuses on holistic narrative understanding on full-length movies, requiring models to reason about the story’s core elements, unlike previous benchmarks that emphasize “needle-in-a-haystack” details. This work seems to pioneer a new area to deal with holistic understanding and reasoning across the entire narrative in movie-level videos.
- MF² is expected high-quality, human-annotated benchmark dataset constructed with intensive human labors including full-time watching movies.
- The multi-faceted analysis—across models, modalities, reasoning types, and comprehension dimensions—is thorough and convincing. The inclusion of human baselines is particularly valuable. It is good information as a baseline for potential users to choose the proposed benchmark dataset.

**Weaknesses:**

- While minimal-edit fibs are effective and annotators filters ambiguous cases, some may be too obvious or, conversely, too subtle, potentially confusing both humans and models.
- Considering the previous benchmarks, this reviewer do not intend to challenge the current configuration. While the authors directly compares the values from the models and humans presented in Table 3, humans would need to watch the video and subtitles without sound for fair comparison.
- For multi-scene cases, the range of multiples is not provided. I think it would be helpful to show its distribution, if the values are collected.
- “global” claims seems to be new and substantial problems to address the overall understanding of the movie. However, it seems there are only 59 cases. Considering the total number of movies, 53, there are only 1.11 claims per a movie. As a main contribution, it looks too small.
- (Minor) In Figures 3-5, the graph visibility could be improved. Rather than simply distinguishing by color, adding another method would make it more visible in grayscale printout.

**Questions:**

Please see weaknesses.

---

> ### Author Response · Authors · 2025-11-20
> **Answer to reviewer BnV2**
>
> We thank the reviewer for their time spent in assessing our work and for recognizing the substantial manual annotation effort behind our benchmark.
>
> >While minimal-edit fibs are effective and annotators filter ambiguous cases, some may be too obvious or, conversely, too subtle, potentially confusing both humans and models.
>
> While we acknowledge that a few claims could, in rare cases, be too obvious or subtle, we emphasize that **quality was a primary design goal**: all claims were manually constructed and carefully reviewed, rather than automatically generated via LLMs for quantity, as usually done in other works. Moreover, our **human evaluation covers all samples**, and humans perform exceptionally well on the task, suggesting that the claims are **clear and not misleading**(see human performance in Table 3). This indicates that model failures are **unlikely to stem from poorly constructed claims**. We also refer the reviewer to the **examples provided in the updated version of the paper (Appendix F, L1568-2089)** for clarity.
>
> >Considering the previous benchmarks, this reviewer do not intend to challenge the current configuration. While the authors directly compare the values from the models and humans presented in Table 3, humans would need to watch the video and subtitles without sound for fair comparison.
>
> We note that our evaluation configuration (using video with subtitles but no audio) is **consistent with the standard practice in most long-form video understanding benchmarks**, as very few models currently support all modalities. Developing models that can effectively process video, audio, and text simultaneously remains an **open challenge**, especially in such long contexts. For instance, Qwen3-Omni-30B-A3B-Instruct requires at least 140GB of GPU memory to process a 120-second video with audio (according to their github). Given that our movies range from 50 to 170 minutes, extending such multimodal processing to our setting would require substantial additional engineering and is far from straightforward. **Importantly, we release the full data, including the original audio tracks, ensuring that our setup does not restrict any future fair comparisons with models that can leverage additional modalities.**
>
> >For multi-scene cases, the range of multiples is not provided. I think it would be helpful to show its distribution, if the values are collected.
>
> We thank the reviewer for this suggestion. While providing the precise range or distribution of scenes for multi-scene claims could be informative and helpful for further analyses, **annotating exact timeframes for every claim would dramatically increase the annotation effort**, given the length of the videos and the number of claims. For context, each annotator already spent **1–2 hours watching a single film and an additional 2–3 hours constructing the corresponding claims** (excluding the multiple rounds of internal discussions and revisions). Collecting precise timeframes across for an entire movie would **at least double this effort**, as half of the claims require reasoning that spans multiple scenes rather than a single localized moment. Therefore, we do not currently have timeframes available for any of the claims, but we are considering including such timeframes in future extensions to further enrich the benchmark.
>
> >“global” claims seems to be new and substantial problems to address the overall understanding of the movie. However, it seems there are only 59 cases. Considering the total number of movies, 53, there are only 1.11 claims per a movie. As a main contribution, it looks too small.
>
> We acknowledge that the number of global claims is smaller than the single-scene and multi-scene categories. Although their number reflects the fact that **constructing global claims is particularly labor-intensive**, as each must be grounded across the entire movie and cannot be answered by reasoning over only three or four (multi) scenes. Nonetheless, global claims are not positioned as the sole or primary contribution of MF2. Instead, they form one component of the benchmark. We consider all claims (single-scene, multi-scene, and global) **equally important**, as they are constructed with the same guiding principle: that **narrative-central events should be memorable to humans** (without rewatching the movie) and therefore suitable for evaluating genuine narrative understanding. Each granularity probes a different type of reasoning, but all share this focus on memorable, narrative-central content, **an angle that distinguishes our benchmark from prior long-video datasets.**
>
> **Closing Note:**
> We thank the reviewer once again for the feedback provided. We hope that our clarifications and replies address your concerns, and we would greatly appreciate any further feedback or reconsideration. We remain open to any additional questions or discussion.

---

> ### Author Response · Authors · 2025-11-25
>
> Dear reviewer, as the discussion period ends in about a week (December 3rd), we are following up to check whether our explanations addressed your comments. We are open to further discussion in the remaining rebuttal period

---

### Official Review · Reviewer_J9hv · 2025-11-06

**Soundness:** 3
**Presentation:** 3
**Contribution:** 2
**Rating:** 4
**Confidence:** 4

**Summary:**

This paper introduces MF2, a benchmark for evaluating long-form movie understanding in vision-language models. It contains 53 open-licensed movies and 868 fact–fib pairs testing reasoning over causal, emotional, temporal, and event aspects. Using a binary claim evaluation protocol, MF2 avoids multiple-choice biases and better measures narrative reasoning. Experiments show that state-of-the-art models (e.g., GPT-4o, Gemini 2.5 Pro) perform well below humans, especially in global and emotional reasoning, revealing current limitations in long-term narrative understanding.

**Strengths:**

1. MF2 introduces a new evaluation paradigm for long-form narrative video understanding, moving beyond prior benchmarks that focus on short-term visual recall or detail-oriented retrieval. It provides a more realistic and cognitively demanding test of narrative comprehension.

2. All samples are manually constructed by the research team after watching entire movies, ensuring annotation quality and consistency. The dataset covers multiple reasoning granularities (single-scene, multi-scene, global) and diverse comprehension dimensions (event, temporal, causal, and emotional understanding), offering a comprehensive assessment of model capabilities.

**Weaknesses:**

- **Overstated Novelty Compared to Existing Long-Video Benchmarks:** While MF2 leverages full-length movies to evaluate long-term reasoning, there already exist several general long-video understanding benchmarks. The authors’ claim that prior works focus only on "peripheral or low-level details" and lack "abstractive understanding of the central storyline" is somewhat overstated. For instance, HourVideo also involves hour-long videos and tasks such as causal and counterfactual reasoning, which similarly require comprehensive narrative understanding.

- **Limited Movie Diversity:** The dataset mainly consists of public-domain films from 1920–1970, which ensures openness but results in relatively narrow thematic and stylistic diversity. This may limit the benchmark’s generalizability to modern, culturally diverse, or visually complex narratives found in contemporary media.

**Questions:**

Please refer to the weakness part.

---

> ### Author Response · Authors · 2025-11-20
> **Answer to reviewer j9hv (part 1)**
>
> We thank the reviewer for their time and effort, and for recognizing the substantial manual annotation, which we consider key to ensuring the benchmark’s quality and reliability.
>
> > The authors’ claim that prior works focus only on "peripheral or low-level details" and lack "abstractive understanding of the central storyline" is somewhat overstated. For instance, HourVideo [...]
>
> We thank the reviewer for raising this point, which helps clarify how our benchmark differs from prior work. Most long-form video benchmarks (>30 minutes), such as LongVideoBench and LVBench, emphasize object recall or localized events rather than deeper narrative comprehension. Although some benchmarks approach tasks related to storyline understanding, our work focuses on a different facet of this problem:**narrative-central events should be memorable, enabling humans to answer related questions without rewatching**. In MF2, indeed human participants perform exceptionally well without revisiting the movie, confirming that our questions target core narrative elements.
>
> Building on this distinction, our inspection of HourVideo’s samples indicates that a large portion of its questions (~80% including temporal frequencies/duration, spatial, information retrieval) focus on details **unlikely to be retained after a single viewing**, particularly given the length and density of the videos (see examples below). While the narrative-related portion (causal, counterfactual, summarization, predictive) constitutes ~10% of the dataset, highlighting this substantial design difference compared to our work (and what each work targets). **This contrast—memorable, narrative-central understanding vs. detailed-oriented recall over long-term videos—is central to our benchmark’s design**.  **We do not to claim that existing datasets completely lack** elements of narrative understanding, but rather **that this is not their primary design focus** and **that our benchmark approaches the problem from a different angle. We clarified this point in the introduction of the updated version (L052-075) to make this distinction explicit and prevent any possible overstatement of novelty.**
>
> Finally, **three distinctions further set our benchmark apart**: (i) manual construction prioritizing quality over quantity, (ii) longer videos (~2× HourVideo), (iii) full human evaluation coverage (vs. 5% in HourVideo), (iiii) and the completely different type of videos (HourVideo focuses on egocentric ones). We will also make these differences explicitly clear in our related work section.
>
> We also encourage the reviewer to consult the examples provided in the **updated version of the paper (Appendix F)**, which illustrate how our claims target memorable narrative-central events. We believe these examples help clarify how our questions differ from the majority of those included in HourVideo. We also illustrate some of the HourVideo ones below. While the following questions may require more than single-frame recall and thus appear to involve long-term understanding, they often resemble “multiple-needles-in-a-haystack” retrieval, which is conceptually different from the narrative-level reasoning that MF2 is designed to assess.
>
>
> Examples from HourVideo:
>
> (Temporal-frequency)
>
> Select the correct statement regarding frequencies of different tool usage in the video
>
> A. The blender jar and the kettle were used with the same frequency by the camera wearer.
>
> B. The tap was used to wash items more frequently than to wash hands during kitchen activities by the camera wearer.
>
> C. The frying pan was not used, only the cooking oil bottle, during the cooking activity.
>
> D. The phone was used more frequently while exercising compared to preparing a drink.
>
> E. Both the egg and the salt were used with the same frequency during egg preparation.
>
> (Spatial)
>
> Select the correct statement regarding the spatial proximity of objects in the video.
>
> A) The camera wearer's seat is equidistant from both the driver's seat and the bus door on the bus.
>
> B) The cashier is closer to the dining table where the camera wearer eats pizza than the trash bin.
>
> C) The driver's seat is positioned directly across from the camera wearer's seat, while the bus door is behind the camera wearer.
>
> D) The weighing station is adjacent to the entrance, with the banana section at the far end.
>
> E) The entrance is nearer to the weighing station than the banana section at the store.

---

> ### Author Response · Authors · 2025-11-20
> **Answer to reviewer j9hv (part 2)**
>
> >Limited Movie Diversity: The dataset mainly consists of public-domain films from 1920–1970, which ensures openness but results in relatively narrow thematic and stylistic diversity. [..]
>
> The decision to use public-domain films was a **trade-off to ensure legal openness, reproducibility, and redistribution**.  While these films reflect the stylistic norms of their historical period, they span a wide **range of genres across five decades**, which does not necessarily imply cultural poverty or homogeneity despite their earlier era. We consider this trade-off **acceptable because fully accessible content avoids the reproducibility issues of datasets** relying on YouTube links or keyframes (e.g., Video-MME, LVBench), which often **become unavailable over time**. Although these films follow predominantly Hollywood-standard narrative structures [1,2,3], current systems still fall far below human performance, highlighting the **benchmark’s difficulty despite reduced stylistic variation**.
>
> Importantly, our goal is **not** to claim that solving this benchmark would amount to solving video reasoning as a whole. Rather, we argue that the skills evaluated here are **necessary but not sufficient** for a model to qualify as a good video reasoning system. We are not suggesting that a model that performs well on MF2 has mastered video reasoning; instead, the substantial gap between human and model performance reveals a clear weakness in current systems (and also a gap in current evaluation frameworks). A human with little exposure to old films can still answer these questions with high accuracy, whereas current models struggle significantly. **These core narrative-understanding abilities remain essential regardless of a film’s era or stylistic traits.**
>
> Stylistic diversity becomes a concern only if the benchmark is interpreted as a **sufficient test of video reasoning, which we do not claim.**
>
> [1]Cutting,J.E.Narrative theory and the dynamics of popular movies. Psychonomic Bulletin & Review,2016.
>
> [2]Pavis, P. Dictionary of the Theatre:Terms, Concepts, and Analysis.University of Toronto Press,1998.
>
> [3]Freytag, G. Freytag’s Technique of the Drama.1896.
>
>
>
> **Closing Note**: We thank the reviewer once again for the feedback. We sincerely hope these points clarify your concerns and will allow you to revisit your score. We welcome any further questions or suggestions.

---

> > ### Author Response · Authors · 2025-11-25
> >
> > Dear reviewer, as the discussion period ends in about a week (December 3rd), we are following up to check whether our clarifications addressed your concerns. We would appreciate your engagement in further discussion or a reconsideration of your assessment if the new information resolves the issues you raised.

---

### Author Response · Authors · 2025-11-20

We would like to thank all reviewers for the time and effort dedicated to review our work. We appreciate that several reviewers highlighted important strengths of our benchmark—particularly the focus on genuine narrative understanding over shallow recall (j9hv, bnv2), the high-quality fully manual annotations (j9hv,bnv2, nok5,kda2), and the inclusion of a human baseline (bnv2,nok5).
We view it as important for the broader discussion to show you examples of our claims. Therefore, we **include in the updated version of the paper (Appendix F, L1568-2089)** a set of examples which illustrate the necessity of visual information in order to be resolved correctly. We also provide Gemini’s predictions (best performing model) along with its explanations.

We also provide one of these examples here:

**Fact (Ground-truth label: True)**:  The grandfather, Peter Benson, intentionally spilled the water on the floor.

**Fib(Ground-truth label: False)**: The grandfather, Peter Benson, accidentally spilled the water on the floor.

Context: Early in the film, the TV breaks. Later, when the criminals have already broken into the house, pop (the grandfather) orchestrates the setup to save everyone (without explicitly stating it). He feigns a heart attack and asks for water, Pidge brings it. The TV technician pretends to repair the broken TV as part of a sabotage plan, while the circuit. When the moment comes, Pop intentionally spills the water on the floor (placing it on the edge of the table). Shortly after, the electrocution occurs when one of the criminals steps on the spilled water. **Both visual and textual information are needed to understand that Pop indeed spilled the water and what was his intention. Gemini fails to predict both claims correctly in every modality setting (see example Tables 8,9 in Appendix F)**.

---

### Author Response · Authors · 2025-12-03
**Discussion Summary for the AC**

Dear AC,

We summarize the main discussion points and our clarifications, organized by reviewer, to assist your assessment. Reviewers highlighted the **fully manual annotation** and **high quality** of the constructed claims (J9hv, BnV2, Nok5, rdkg, KDA2), as well as the dataset’s **legal openness** (Nok5). BnV2 noted the thoroughness of our evaluation, and KDA2 highlighted that MF² avoids model-bias issues common in semi-automatically generated benchmarks.


**Reviewer J9hv**

Clarifications provided:

- Regarding reviewer's comment about novelty: apart from noting differences from benchmarks such as HourVideo (specifically mentioned by the reviewer), we clarified that MF²  targets memorable, narrative-central events rather than detail-oriented recall. **This distinction is made explicit in the updated introduction (L052–075)**, along with the **examples illustrating that our claims target memorable narrative-central events** (Appendix F).

- Regarding stylistic/thematic diversity of movies: we prioritized legal openness at the cost of some stylistic breadth, though the 50-year span still provides considerable variation. Importantly, a human with little exposure to old films can still answer these questions with high accuracy, whereas current models struggle significantly. **The core narrative-understanding abilities remain essential regardless of a film’s era or stylistic traits.**



**Reviewer BnV2**

Clarifications provided:

- Some claims might be “too obvious or too subtle”: we clarified that **humans perform exceptionally well on the task, indicating that the claims are clear** and model failures are unlikely due to poor construction. We additionally provided examples of claims (App F).

- Model vs human baseline comparison: we clarified that evaluating models on video with subtitles (excluding audio) is a **standard evaluation practice** in long-video benchmarks, and we **release all data** (including audio) for future comparisons including all modalities.



**Reviewer Nok5**

Experiments & Clarifications provided:

- Variety of models: Following the reviewer’s suggestion, we **added more VLMs** to the evaluation suite (App D).

- Regarding stylistic/thematic diversity of movies: we clarified that we prioritized legal openness at the cost of some stylistic breadth, though the 50-year span still provides considerable variation. Importantly, a human with little exposure to old films can still answer these questions with high accuracy, whereas current models struggle significantly. **The core narrative-understanding abilities remain essential regardless of a film’s era or stylistic traits.**

- “All annotators are co-authors”: we explained thoroughly the rationale behind this choice. This practice is also aligned with established benchmarks.



**Reviewer rdkg**

Experiments & Clarifications provided:

- Claims requiring visual information: we **conducted ablations according to reviewer’s suggestions**, including subtitles-only for two open-weight models (rebuttal), frame-shuffling (App E), and failure case examples (App F).

- We clarified the choice of pairwise vs. simple accuracy and **presented results** (rebuttal) showing why simple accuracy is insufficient; **we also included requested experiments** with confidence scores (rebuttal).

- We provided **thorough responses to all points** (see corresponding discussion); for brevity, we summarized only the key clarifications here.



**Reviewer KDA2**

Experiments & Clarifications provided:

- We clarified the choice of pairwise vs. simple accuracy and **presented results** showing why simple accuracy is insufficient (rebuttal).

- Regarding stylistic/thematic diversity of movies: we prioritized legal openness at the cost of some stylistic breadth, though the 50-year span still provides considerable variation. Importantly, a human with little exposure to old films can still answer these questions with high accuracy, whereas current models struggle significantly. **The core narrative-understanding abilities remain essential regardless of a film’s era or stylistic traits.**

- We clarified **differences from Trope datasets and highlighted benchmark novelty**.

Outcome: The reviewer acknowledged the clarifications and additional results, and raised the score from 6 to 8 (22 Nov).


**We thank you for your time** and hope this summary will assist your assessment.

---

### Meta-Review · Area_Chair_Bxpp · 2026-01-07

**Summary:**

This paper introduces MF2, a benchmark for long-form movie understanding in vision-language models (VLMs). It includes 50 open-licensed films and 850 fact–fib pairs designed to evaluate causal, temporal, emotional, and event-level reasoning. Experiments demonstrate that current SOTA VLMs lag significantly behind human performance, particularly in global and emotional understanding.

Reviewers agree that MF2 targets an important and underexplored capability of VLMs. The use of open-licensed movies, human-annotated claims, and human baselines supports reproducibility and strong evaluation quality, making the benchmark potentially valuable to the community.

However, several concerns remain, particularly regarding overstated novelty, limited data diversity, and possible annotation bias.
Some reviewers felt that the paper overstates its novelty relative to existing long-video benchmarks. The public-domain films are from 1920–1970, which raises concerns about thematic and visual diversity and limits generalizability to modern content. In addition, the fact that all annotations were produced by co-authors introduces potential bias, which was not fully addressed in the rebuttal.

Overall, while the paper presents a useful benchmark and tackles an important problem, the remaining concerns slightly outweigh its strengths. I therefore recommend rejection at this time. Clarifying distinctions from prior benchmarks and expanding both data and annotation diversity would strengthen a future submission.

**Reviewer Concerns:**

- Reviewer J9hv: Remaining concerns focus on overstated novelty and limited data diversity.

- Most concerns raised by Reviewers Nok5 and rdkg were addressed. Remaining issues include limited data diversity and potential annotation bias. Since all annotations were produced by co-authors, this may introduce subjectivity in claim design and quality control. The rebuttal did not sufficiently clarify the diversity of annotator backgrounds (e.g., domain expertise, cultural diversity, language proficiency) or explain how such diversity mitigates these biases.

- Reviewers BnV2 and KDA2: Concerns were well resolved.

**Reviewer Scores:**

- Reviewer J9hv: Score remains 4; concerns about overstated novelty and limited diversity persist.

- Reviewer Nok5: Score remains 4; concerns about limited diversity and annotator bias remain.

- Reviewer rdkg: Score remains 4; most concerns addressed except for data diversity.

- Reviewer BnV2: Score 6; concerns resolved.

- Reviewer KDA2: Score increased from 6 -> 8; most concerns addressed.

---

### Decision · Program_Chairs · 2026-01-26

Reject